# A Call to Lagrangian Action:
# Learning Population Mechanics from Temporal Snapshots

**Vincent Guan** [1 2]   **Lazar Atanackovic** [3 4 5]   **Kirill Neklyudov** [2 6 7]

## Abstract

The population dynamics of molecules, cells, and organisms are governed by a number of unknown forces. In the last decade, population dynamics have predominantly been modeled with Wasserstein gradient flows. However, since gradient flows minimize free energy, they fail to capture important dynamical properties, such as periodicity. In this work, we propose a change in perspective by considering dynamics that minimize a *population-level action* under a damped Wasserstein Lagrangian. By deriving the corresponding Hamiltonian equations of motion, we formalize *Wasserstein Lagrangian Mechanics*, a structured class of second-order dynamics that encompasses classical mechanics, quantum mechanics, and gradient flows. We then propose `WLM` as the first algorithm that learns these second-order dynamics from observed marginals, without specifying the Lagrangian. By directly learning the population mechanics, `WLM` can both forecast and interpolate unseen marginals, and outperforms existing gradient flow and flow matching methods across a wide range of dynamics, including vortex dynamics, embryonic development, and flocking.

## 1. Introduction

Population dynamics govern every level of nature, from the Brownian motion of molecules in a fluid, to the embryonic differentiation of developing cells, to the swarm behaviour of animals (May, 1987; Codling et al., 2008). Identifying the underlying forces that drive the evolution of a given population is therefore a fundamental, but difficult problem (Newman et al., 2014; Agozzino et al., 2020).

[1]University of British Columbia [2]Mila - Quebec AI Institute [3]University of Alberta [4]Alberta Machine Intelligence Institute [5]Broad Institute of MIT and Harvard [6]Université de Montréal [7]Institut Courtois. Correspondence to: Vincent Guan <vincentguan23@gmail.com>, Kirill Neklyudov <k.necludov@gmail.com>.

*Proceedings of the 43rd International Conference on Machine Learning*, Seoul, South Korea. PMLR 306, 2026. Copyright 2026 by the author(s).

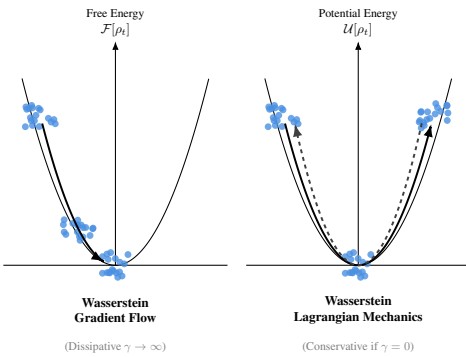

*Figure 1.* Wasserstein gradient flows describe first-order population dynamics that minimize the free energy $\mathcal{F}[\rho_t]$. We propose Wasserstein Lagrangian mechanics (WLM), which describe a richer class of damped second-order dynamics, based on the population-level potential energy $\mathcal{U}[\rho_t]$. Given the same quadratic functional, gradient flows dissipate until equilibrium, while WLM produces oscillating dynamics if $\gamma = 0$, and degenerate to gradient flow dynamics in the overdamped limit $\gamma \to \infty$.

If domain knowledge is available, then a prior can be prescribed on the dynamics (Shen et al., 2024; Simpson & Plank, 2025), but in most cases, we are unable to obtain an accurate reference process. The predominant approach in machine learning and computational biology has been to model population dynamics as a gradient flow (Hashimoto et al., 2016; Weinreb et al., 2018; Lavenant et al., 2021; Bunne et al., 2022; Terpin et al., 2024; Guan et al., 2026). Indeed, gradient flows have a rich mathematical foundation (Ambrosio et al., 2008) and admit optimization schemes for tractable inference (Jordan et al., 1998). However, since gradient flows minimize free energy, they can only characterize purely dissipative aperiodic dynamics. Flow-based methods (Atanackovic et al., 2024; Kapusniak et al., 2024; Petrović et al., 2025) have been developed to address these limitations and produce higher quality interpolations, but they cannot forecast beyond the observed horizon.

A change of perspective is therefore necessary to model more general population dynamics and to forecast future dynamics. As a natural starting point, we consider *the principle of least Wasserstein action*:

> The population marginals $(p_t)_{0 \leq t \leq T}$ minimize the action of some population-level Lagrangian $\mathcal{L}[\rho_t, \dot{\rho}_t, t]$.

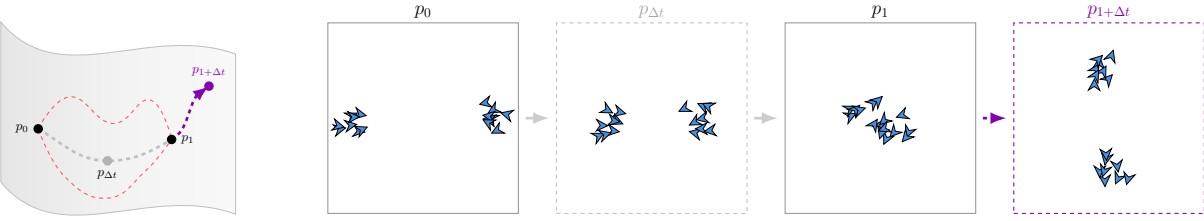

**(a) Wasserstein Lagrangian flow in $\mathcal{P}(\mathbb{R}^d)$**   **(b) Wasserstein Hamiltonian mechanics (visualized in $\mathbb{R}^d$)**

*Figure 2.* **Learning population mechanics with WLM:** In **(a)**, we illustrate the principle of least Wasserstein action (3): given observed marginals $p_0$ and $p_1$, the true interpolants form a minimal action curve in the space of densities, with respect to a population-level Lagrangian action. Alternative curves of densities have higher action and are drawn in red. Wasserstein least action induces Hamiltonian mechanics on the population, which we visualize in the state space for a population of interacting Boids in **(b)**. Our method (WLM) learns these mechanics to interpolate and forecast unseen marginals (dashed grey and violet panels).

Least action in the Wasserstein space of probability distributions has been proposed in recent literature (Chow et al., 2020; Neklyudov et al., 2023a), and states that the population evolves according to the most efficient path, as determined by the population-level Lagrangian. By defining the Lagrangian on the Wasserstein space, rather than the state space, this formulation captures emergent dynamics. For example, the evolution could be determined by interactions between individuals, as well as the population's intrinsic properties, like its central mass. Moreover, by working directly on the space of marginals, this framework is directly amenable to settings where it is impossible to track trajectories, such as single-cell biology. While least action is intuitively attractive, gradient flow and flow matching remain the prevailing machine learning approaches for learning population dynamics. To motivate least action as an advantageous framework, two key questions must be answered:

1. Does Wasserstein least action capture more expressive dynamics than Wasserstein gradient flows?

2. How can least action dynamics be inferred from observational data, without specifying the true Lagrangian?

**Contributions.**   In this work, we propose *Wasserstein Lagrangian Mechanics* (WLM), which *re-frames learning population dynamics as learning the population mechanics of an evolving population.* We prove that WLM is significantly more expressive than Wasserstein gradient flows (see Figure 1), and complement our theory by developing an algorithm that learns the mechanics directly from data. We outline our contributions and the structure of our paper:

- **Theory:** In Section 2, we formalize WLM by deriving the Hamiltonian equations of motion from the least action principle under a damped Wasserstein Lagrangian. These equations of motion describe a structured class of population mechanics, which encompasses classical mechanics, quantum mechanics, and gradient flows.

- **Method:** In Section 3, we propose WLM, a neural mechanics model that learns population mechanics directly from data. WLM learns the system's Hamiltonian mechanics by parameterizing the population-level potential energy and training the model to predict its next $k$ marginals.

- **Experiments:** In Section 4, we demonstrate WLM's ability to learn gradient flows, curling dynamics from ocean vortex data, cell dynamics from scRNA data, and emergent dynamics from Boids. By learning the population mechanics, WLM consistently outperforms state-of-the-art methods for both *forecasting* and *interpolation*, and does so without requiring a reference process.

**Conflict of Interest Disclosure.**   The authors declare no financial conflicts of interest.

## 2. Wasserstein Lagrangian Mechanics

In this section, we develop a mathematical framework for the mechanics of an evolving population. We first review population dynamics under the canonical continuity equation, and define population-level coordinates for flows that minimize a Wasserstein Lagrangian action. Then, from the Wasserstein least action principle, we determine the *population mechanics* by deriving the Hamiltonian equations of motion. This characterizes the population as a damped second-order system, driven by the population-level potential energy $\mathcal{U}[\rho_t]$. Finally, we use these insights to clarify the scope of dynamics that WLM encompasses, including gradient flows, as well as classical and quantum mechanics.

### 2.1. Continuous Population Dynamics

We recall an important theorem from Ambrosio et al. (2008): continuous population dynamics in the Wasserstein-2 space $\mathcal{P}_2(\mathbb{R}^d)$ can always be represented by the continuity equation, with time-varying vector field $v_t = \nabla s_t$.

**Proposition 2.1** (Ambrosio et al. (2008, Theorem 8.3.1)). *Let $(\rho_t)_{0 \leq t \leq 1}$ be an absolutely continuous curve of measures in $\mathcal{P}_2(\mathbb{R}^d)$. Then, there exists a unique (up to a constant) potential $s_t : \mathbb{R}^d \to \mathbb{R}$ such that the population*

*dynamics* $(\rho_t)_{0 \leq t \leq 1}$ *are defined by the evolution*

$$\dot{\rho}_t = -\nabla \cdot (\rho_t \nabla s_t). \tag{1}$$

*Moreover, of all vector fields $v_t$, which generate $(\rho_t)_{0 \leq t \leq 1}$ via $\dot{\rho}_t = -\nabla \cdot (\rho_t v_t)$, $\nabla s_t$ is the one that minimizes the $W_2$ kinetic energy for a.e. $t \in [0,1]$:*

$$\mathcal{K}[\rho_t, \dot{\rho}_t] = \frac{1}{2} \int \|v_t\|^2 \rho_t dx. \tag{2}$$

Proposition 2.1 offers a canonical representation for continuous population dynamics, but we note that population marginals can be produced by multiple laws on paths, unless identifiability conditions hold (Guan et al., 2024; 2026). In this work, we assume that the population dynamics are absolutely continuous in $\mathcal{P}_2(\mathbb{R}^d)$ and use the canonical representation (1). This allows us to conveniently rewrite the coordinates $(\rho_t, \dot{\rho}_t)$ as $(\rho_t, s_t)$, such that $\rho_t \in \mathcal{P}_2(\mathbb{R}^d)$ defines (Wasserstein) position and $\nabla s_t$ is a tangent vector (Ambrosio et al., 2008)[Section 8.4].

### 2.2. Wasserstein Least Action Population Dynamics

We now consider population dynamics that minimize a Wasserstein Lagrangian action (Neklyudov et al., 2023b). As in classical mechanics, the simplest Wasserstein Lagrangian is the sum of the kinetic and (negative) potential energy functionals. The scalar energy values are defined at the population level, via the Wasserstein coordinates $(\rho_t, s_t)$:

$$\mathcal{L}[\rho_t, s_t] = \mathcal{K}[\rho_t, s_t] - \mathcal{U}[\rho_t].$$

In this work, we fix $\mathcal{K}[\rho_t, s_t]$ as the $W_2$ kinetic energy (2) with the canonical vector field $v_t = \nabla s_t$. The potential energy $\mathcal{U}[\rho_t]$ is the more crucial and flexible term, as it can be any functional of the density, including entropy, interaction kernels, and expectations. We also note that more complicated Lagrangians can explicitly depend on time.

If we fix a Lagrangian $\mathcal{L}[\rho_t, s_t, t]$, and a set of marginals $\{p_{t_i}\}_{i=1}^{M}$, then the most efficient curve of marginals with respect to the Lagrangian action is the one that obeys

$$(p_t)_{0 \leq t \leq 1} = \arg \min_{\substack{\{\rho_t : [0,1] \to \mathcal{P}_2(\mathbb{R}^d) \\ \text{s.t. } \rho_{t_i} = p_{t_i} \forall i=1,\dots,M\}}} \mathcal{S}[\rho_t, s_t, t] \tag{3}$$

$$\mathcal{S}[\rho_t, s_t, t] = \int_0^1 \mathcal{L}[\rho_t, s_t, t] dt.$$

We can also characterize the least action dynamics $(p_t)_{0 \leq t \leq 1}$ as having coordinates $(\rho_t, s_t)$ that obey the first-order conditions

$$\frac{\delta \mathcal{S}[\rho_t, s_t]}{\delta \rho_t}[h] = \frac{\delta \mathcal{S}[\rho_t, s_t]}{\delta s_t}[h] = 0, \tag{4}$$

which holds for all test functions $h \in C_c^\infty(\mathbb{R}^d \times (0,1))$. Note that this matches classical stationary action, but replaces pointwise derivatives with functional derivatives, since position and velocity are defined on $\mathcal{P}_2(\mathbb{R}^d)$.

### 2.3. Hamiltonian Equations of Motion

We have seen that without loss of generality, population dynamics evolve according to the continuity equation (1) for some time-varying vector field $\nabla s_t$. A natural question to ask is if we can determine insights about the driving potential $s_t$. We can in fact derive its evolution equation under the principle of least Wasserstein action.

**Theorem 2.2** (Hamiltonian equations of motion). *Let $(p_t)_{0 \leq t \leq 1}$ minimize action with respect to the damped Wasserstein Lagrangian*

$$\mathcal{L}[\rho_t, s_t, t] = e^{\gamma t} \left( \frac{1}{2} \int \|\nabla s_t\|^2 \rho_t dx - \mathcal{U}[\rho_t] \right) \tag{5}$$

*with damping $\gamma \geq 0$*

*Then, $(p_t)_{0 \leq t \leq 1}$ obeys the continuity equation (1), such that the driving potential $s_t$ obeys*

$$\dot{s}_t(x) = -\frac{1}{2}\|\nabla s_t(x)\|^2 - \frac{\delta \mathcal{U}[\rho_t]}{\delta \rho_t}(x) - \gamma s_t(x). \tag{6}$$

*The marginals $(p_t)_{0 \leq t \leq 1}$ can therefore be produced by sampling trajectories, obeying the second-order mechanics:*

$$\frac{d}{dt} x_t := v_t := \nabla s_t(x_t), \qquad\qquad x_0 \sim p_0 \tag{7}$$

$$\frac{d}{dt} v_t = -\nabla \frac{\delta \mathcal{U}[\rho_t]}{\delta \rho_t}(x_t) - \gamma v_t, \quad v_0 = \nabla s_0(x_0) \tag{8}$$

*Proof.* We present the full derivation in Appendix A. □

The Hamiltonian equations of motion (8) offer a complementary mechanical perspective to the principle of least Wasserstein action (3), which can be visualized in the state space (see Figure 2). Intuitively, the population obeys second-order mechanics under a 'generalized Newton's law': individual particles are accelerated by a force determined by the population-level potential energy $\mathcal{U}[\rho_t]$. This yields an elegant interpretation: under the principle of least action, the population drives itself, by using the potential energy of its current configuration to evolve its transport vector field $\nabla s_t$. While these mechanics produce the true marginals at each time, we note that multiple laws on paths explain the same population dynamics (see Appendix B).

We note that similar Wasserstein Hamiltonian equations have been derived in the conservative regime (Chow et al., 2020, Proposition 2). By additionally considering the damping parameter $\gamma \geq 0$, we characterize a more expressive class of dissipative population mechanics, which includes gradient flows, as we will see in the next section.

## 2.4. Expressivity of Dynamics under WLM

In this section, we show that the class of damped second-order dynamics described by Wasserstein Lagrangian Mechanics encompasses a range of fundamental dynamics. First, we review the undamped setting, which characterizes classical and quantum mechanics.

**Proposition 2.3** (WLM describes classical mechanics). *If $\gamma = 0$ and $\mathcal{U}[\rho_t] = \int U(x)\rho_t(x)dx$, then WLM describes Newtonian mechanics.*

*Proof.* Under Newtonian mechanics, a particle's acceleration follows the gradient of a state-space potential energy, $\nabla U(x)$ (Arnold et al., 1989). We thus recover Newtonian mechanics from the Hamiltonian equations (8) if $\gamma = 0$ and the Wasserstein potential energy is an expectation. Indeed, $\mathcal{U}[\rho_t] = \int U(x)\rho_t(x)dx \implies \frac{\delta\mathcal{U}[\rho_t]}{\delta\rho_t}(x) = U(x)$. $\square$

In other words, if $\mathcal{U}[\rho_t]$ acts linearly on the density via the state-space potential $U(x)$, then individual masses would evolve independently of the rest of the population. These functionals describe classical mechanics, including gravitational motion, harmonic oscillation, and rigid body rotation.

By adding a nonlinear Fisher information term to the potential energy functional, WLM also describes the Schrödinger equation, which governs the wave function in quantum mechanics (see Chow et al. (2020, Example 3) and Neklyudov et al. (2023b, Section B.3)).

We now show that Wasserstein gradient flows admit two characterizations under WLM: a stable characterization in the limit of overdamped friction, and an unstable characterization as a conservative system, which requires a specific initial velocity.

**Proposition 2.4** (WLM describes gradient flows). *Let $(\rho_t)_{t\geq 0}$ minimize a free energy functional $\mathcal{F}[\rho]$. Then the dynamics $(\rho_t)_{t\geq 0}$ can be described by WLM as*

1. *The overdamped limit ($\gamma \to \infty$) of Wasserstein Lagrangian systems with $\mathcal{U}[\rho_t] = \mathcal{F}[\rho_t]$.*
2. *A conservative system ($\gamma = 0$) with 'inverted' potential energy $\mathcal{U}[\rho_t] = -\frac{1}{2}\int \|\nabla\frac{\delta}{\delta\rho_t}\mathcal{F}[\rho_t]\|^2\rho_t dx$ and initial velocity field $v_0 \propto -\nabla\frac{\delta}{\delta\rho_0}\mathcal{F}[\rho_0]$.*

*Proof.* To recover the interpretation of a gradient flow as the overdamped limit of Lagrangian systems (see Villani et al. (2009, Chapter 7) and Adams et al. (2013)), but generalized to the Wasserstein space, we first recall that given the canonical coordinates, a gradient flow of $\mathcal{F}$ has vector field proportional to the Wasserstein gradient of the free energy. Formally, $\nabla s_t \propto -\nabla\frac{\delta}{\delta\rho_t}\mathcal{F}[\rho_t]$ holds $\rho_t$-a.e. (Santambrogio, 2017, (4.10)). Then, we consider the Hamiltonian equations (8) with $\mathcal{U}[\rho_t] = \mathcal{F}[\rho_t]$ and take $\gamma \to \infty$. We may apply Tikhonov's theorem (Tikhonov, 1952) to obtain

$$\frac{dx}{dt} \propto -\frac{\delta\mathcal{F}[\rho_t]}{\delta\rho_t}(x) \implies \nabla s_t \propto -\nabla\frac{\delta\mathcal{F}[\rho_t]}{\delta\rho_t}.$$

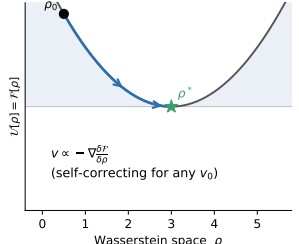 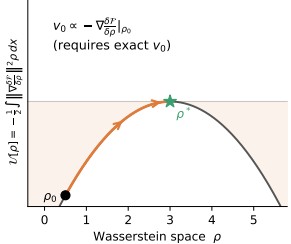

*(a)* Stable overdamped system ($\gamma \to \infty$) with $\mathcal{U}[\rho] = \mathcal{F}[\rho]$

*(b)* Unstable conservative system with inverted potential

*Figure 3.* We visualize Proposition 2.4, which shows that Wasserstein gradient flows admit two characterizations under WLM.

The implication is from the principle of superposition (Ambrosio et al., 2008, Theorem 8.2.1). It follows that the overdamped characterization produces gradient flow dynamics for $t > 0$, no matter what $v_0$ is initialized as.

Then, to prove the second description, we note that since a gradient flow has vector field $\nabla s_t \propto -\nabla\frac{\delta}{\delta\rho_t}\mathcal{F}[\rho_t]$, it follows that, between any marginals $p_0$ and $p_1$, the population dynamics minimize

$$\mathcal{J}[\rho_t, s_t] = \frac{1}{2}\int_0^1 \int \|\nabla s_t + \nabla\frac{\delta}{\delta\rho_t}\mathcal{F}[\rho_t]\|^2\rho_t \, dx \, dt,$$

We then simplify and obtain the equivalent minimization

$$\mathcal{S}[\rho_t, s_t] = \int_0^1 \frac{1}{2}\int \|\nabla s_t\|^2\rho_t + \|\nabla\frac{\delta}{\delta\rho_t}\mathcal{F}[\rho_t]\|^2\rho_t \, dxdt,$$

which is the Lagrangian action with potential energy $\mathcal{U}[\rho_t] = -\frac{1}{2}\int\|\nabla\frac{\delta}{\delta\rho_t}\mathcal{F}[\rho_t]\|^2\rho_t dx$. However, these dynamics only determine a gradient flow if we also have $\nabla s_0 \propto -\nabla\frac{\delta}{\delta\rho_0}\mathcal{F}[\rho_0]$. Otherwise, the gradient flow condition $\nabla s_t \propto -\nabla\frac{\delta}{\delta\rho_t}\mathcal{F}[\rho_t]$ fails to hold for $t > 0$ (see Proposition A.2). $\square$

We offer a visual interpretation of Proposition 2.4 in Figure 3. Intuitively, while gradient flows are an expressive class of dynamics, popularly used for modeling population dynamics, Proposition 2.4 shows that they only represent an extremely specific subset of Wasserstein Lagrangian Mechanics. Indeed, the free energy $\mathcal{F}[\rho]$ and the potential energy $\mathcal{U}[\rho]$ can each be arbitrary functionals, but gradient flows produce first-order mechanics along the free energy landscape determined by $\mathcal{F}[\rho]$, while WLM produces second-order mechanics determined by $\mathcal{U}[\rho]$ and $\gamma$. Given equality $\mathcal{U}[\rho] = \mathcal{F}[\rho]$, the WLM system only degenerates into a gradient flow in the overdamped limit. Similarly, Proposition 2.4 shows that gradient flows can also be represented as highly unstable conservative systems, but only if the initial velocity obeys $v_0 \propto \nabla s_0$. Thus, any deviation in $v_0$ still characterizes a WLM system, but will fail to be a gradient flow of $\mathcal{F}$.

## 3. Learning Population Mechanics

Having demonstrated WLM's expressivity, we now turn to the task of inferring these mechanics from data. We note that if the Lagrangian $\mathcal{L}[\rho_t, s_t, t]$ is known, then the interpolants between observed marginals can be estimated by minimizing the action functional (3) with respect to $\mathcal{L}[\rho_t, s_t, t]$(Neklyudov et al., 2023b; Qian et al., 2024; Sun et al., 2025). However, it is not at all obvious how to pick the correct Lagrangian, since this amounts to choosing a reference process. We therefore consider the Hamiltonian perspective, and seek to learn population mechanics that produce the observed marginals.

**Mechanics Model.** We propose a neural mechanics model to learn the population mechanics from a sequence of empirical marginals $\{\hat{p}_{t_i}\}_{i=1}^M$ and the initial velocities $\hat{v}_0$. The intuition behind our algorithm comes from Theorem 2.2: given the initial conditions, the population dynamics only depend on the potential energy $\mathcal{U}[\rho_t]$ and the damping $\gamma$. We therefore parameterize these variables and estimate them from the observed marginals and the initial velocity $v_0$. The learned mechanical system can then be simulated from any time, and can thus be used for interpolation or forecasting.

First, recall that the acceleration of a particle $x_t^{(j)} \sim p_t$ is given by $\frac{dv_t^{(j)}}{dt} = -\nabla_x \frac{\delta \mathcal{U}[p_t]}{\delta p_t}(x_t^{(j)}) - \gamma v_t^{(j)}$. While the functional derivative is generally intractable, Proposition 3.1 (proof in Appendix) shows that it can be bypassed with a tractable characterization.

*Proposition* 3.1. Consider the empirical measure $\hat{p} = \frac{1}{N}\sum_{i=1}^N \delta_{x^{(i)}}$ and let $\Psi(x^{(1)}, \ldots, x^{(N)}) := \mathcal{U}[\hat{p}]$ be its potential energy. Then, for any particle $x^{(j)} \sim \hat{p}$, we have

$$\nabla_{x^{(j)}}\Psi(x^{(1)}, \ldots, x^{(N)}) = \frac{1}{N}\nabla_x \frac{\delta \mathcal{U}[p]}{\delta p}(x^{(j)})\Big|_{p=\hat{p}} \quad (9)$$

We may therefore parameterize $\Psi_\theta : \mathbb{R}^{N \times d} \to \mathbb{R}$ with a deep neural network and obtain accelerations from the model via automatic differentiation:

$$a(x_t^{(j)}) = -\nabla_{x_t^{(j)}}\Psi_\theta(x_t^{(1)}, \ldots, x_t^{(N)}) - \gamma v_t^{(j)}, \quad (10)$$

where we absorb the factor $N$ into the parameterization.

**Training Algorithm.** To train our model, we access accelerations via (10), and roll out predicted dynamics from the initial conditions $(\hat{p}_0, \hat{v}_0)$ via the second-order Verlet integrator (Leapfrog), due to its stability for Hamiltonian systems (Birdsall & Langdon, 2018). We then set the loss to be the divergence, e.g. Sinkhorn divergence, between the observed marginals, $\{\hat{p}_{t_i}\}_{i=1}^k$, and the model's simulated population. To update the model, we evaluate its gradient with respect to parameters $\theta$ by backpropagating directly through the time-discretized trajectories, i.e. the so-called Discretize-Then-Optimize approach. The pseudo-code for training the model and for simulating the mechanics are

---

**Algorithm 1** Learning Population Mechanics

1: **Input:** Observed snapshots $\{X_{t_i} = \{x_{t_i}^{(j)}\}_{j=1}^N\}_{i=0}^M$, initial velocities $v_0 = \{v_0^{(j)}\}_{j=1}^N$
2: **Hyperparams:** Divergence $\mathcal{D}$, damping $\gamma \geq 0$, step size $\Delta t$
3: **Initialize:** Energy model $\Psi_\theta : \mathbb{R}^{N \times d} \to \mathbb{R}$
4: **for** iter = 1 **to** max-iterations **do**
5:     Sample rollout horizon $K \sim \text{Unif}[1, M]$
6:     $\{\hat{X}_{t_i}\}_{i=1}^K \leftarrow \text{P-Mechanics}(\Psi_\theta, \gamma, X_{t_0}, v_0, \Delta t, K)$
7:     $\mathcal{L} \leftarrow \frac{1}{k}\sum_{i=1}^K \mathcal{D}(\hat{X}_{t_i}, X_{t_i})$
8:     $\theta \leftarrow \text{Optimizer}(\theta, \nabla_\theta \mathcal{L})$
9:     **if** $\gamma$ learnable **then** $\gamma \leftarrow \text{Optimizer}(\gamma, \nabla_\gamma \mathcal{L})$
10: **end for**
11: **Return:** Optimized $\Psi_\theta$ and $\gamma$

---

1: **Input:** $\Psi_\theta, \gamma, \{x_0^{(j)}\}_{j=1}^N, \{v_0^{(j)}\}_{j=1}^N, \Delta t, K$
2: **for** $k = 0$ **to** $K - 1$ **do**
3:     $a_k^{(j)} \leftarrow -\nabla_{x_k^{(j)}}\Psi_\theta(x_k^{(1)}, \ldots, x_k^{(N)})$ for all $j$
4:     $x_{k+1}^{(j)}, v_{k+1}^{(j)} \leftarrow \text{Leapfrog}(x_k^{(j)}, v_k^{(j)}, a_k^{(j)}, \gamma, \Delta t)$ for all $j$
5: **end for**
6: **Return:** $\{\{x_k^{(j)}\}_{j=1}^N\}_{k=0}^K$

---

given in Algorithm 1 and Algorithm 2 respectively. We note that temporal features can be added to model time-varying potential energy $\mathcal{U}[\rho_t, t]$, and that the damping parameter can either be fixed, e.g. $\gamma = 0$, or initialized and learned.

## 4. Experiments

In Section 2.4, we proved that WLM describes classical mechanics, quantum mechanics, and gradient flows. We first evaluate on a benchmark dataset of gradient-flow SDEs to demonstrate that WLM can indeed learn any dynamics modeled by gradient flows. We then consider popular real data settings from the literature to evaluate WLM's performance on non-gradient dynamics. Specifically we consider an ocean vortex dataset (Shen et al., 2024; Petrović et al., 2025; Berlinghieri et al., 2025) and an embryonic scRNA dataset (Tong et al., 2020; 2023b; Neklyudov et al., 2023a; Kapusniak et al., 2024). Finally, we introduce a novel Boids (Reynolds, 1987) dataset to evaluate inference on emergent dynamics from a population of interacting agents.

**Experimental Setup.** We evaluate the ability of methods to *forecast* and *interpolate* by computing the $W_1$ distance between true and predicted marginals. To evaluate *forecasting* quality, we simulate the learned dynamics from $t = 0$, and distinguish the results between seen (train) and unseen (test) marginals. To evaluate *interpolation* quality for an unseen marginal $p_h$, we simulate from the previous observed time $h - 1$, as done in Neklyudov et al. (2023b). This ensures a fairer comparison against flow and SB based models, which are constrained to match the training marginals. To simulate methods that fit an SDE, we use the Euler-Maruyama scheme, and for WLM, we use the leapfrog scheme, as shown in Algorithm 2. At inference time, we use 5 substeps between marginals. We describe

*Table 1.* Gradient flow SDEs: We report the average $W_1$ distances between true marginals and learned marginals rolled out from JKONET*, NN-APPEX, and WLM. We consider both paired and unpaired settings, and we distinguish between train (first 10 marginals) and forecast (next 10 marginals). Results are also averaged across all 5 SDEs. The full breakdown per SDE is in Table 6.

| | Paired | | Unpaired | |
|---|---|---|---|---|
| Method | Train $W_1$ | Forecast $W_1$ | Train $W_1$ | Forecast $W_1$ |
| JKONET* | $0.085 \pm 0.007$ | $0.193 \pm 0.020$ | $0.236 \pm 0.040$ | $1.618 \pm 0.261$ |
| NN-APPEX | $0.080 \pm 0.006$ | $\mathbf{0.131 \pm 0.006}$ | $0.102 \pm 0.008$ | $0.260 \pm 0.025$ |
| WLM (learnable friction) | $\mathbf{0.062 \pm 0.004}$ | $0.137 \pm 0.012$ | $\mathbf{0.068 \pm 0.004}$ | $\mathbf{0.246 \pm 0.026}$ |
| WLM (0 friction) | $0.119 \pm 0.012$ | $0.255 \pm 0.037$ | $0.119 \pm 0.011$ | $0.346 \pm 0.045$ |

the architecture, hyperparameters, and runtimes of WLM in Appendix C. The code repository is available on GitHub: https://github.com/guanton/WLM.

### 4.1. Gradient-flow Dynamics: Potential-driven SDEs

To show that WLM can learn gradient flow dynamics, we obtain SDEs such that the drift is a potential landscape $-\nabla V$ from a benchmark dataset (Terpin et al., 2024; Persiianov et al., 2025; Guan et al., 2026) and we set the diffusivity to be $\sigma^2 = 1$. For each SDE (the exact potentials are defined in Appendix C.1), we simulate 10 evenly spaced marginals with 1000 samples each (training data), and reserve the next 10 marginals for forecasting (test data). We also distinguish between "paired" and "unpaired" data settings. In the paired setting, we evolve the same 1000 particles across all times, while in the unpaired setting, we obtain independent samples for each marginal.

Recall that by Proposition 2.4, gradient flows can be characterized as overdamped systems or as conservative systems with precise initial velocity. We implement both versions for WLM. For the former, we provide the analytic velocity field from the free energy, $v_0(x) = -\nabla(V(x) + \frac{1}{2} \log p_0(x))$ and fix $\gamma = 0$. For the latter, we set $v_0 = 0$ and make friction learnable (initial $\gamma = 1$).

We consider JKONET* (Terpin et al., 2024) and NN-APPEX (Guan et al., 2026) as our gradient flow baselines, due to their performance and ability to estimate both the drift $-\nabla V$ and the diffusion $\sigma^2$ from marginals (Guan et al., 2026).

We show results in Table 1. Every method succeeds at forecasting in the paired data setting, and WLM (with learnable friction) achieves the best performance in the significantly harder unpaired setting. As observed in Persiianov et al. (2025, A.3), JKONET* significantly struggles in the unpaired setting due to its reliance on pre-computed couplings. The results therefore show that by making friction learnable, WLM learns gradient flow dynamics as effectively as existing gradient-flow methods. In fact, these results suggest that the learned dissipation by WLM can be used to determine how close population dynamics are to being a gradient flow. We consistently recover $\gamma \geq 500$ in most runs of this experiment (see Figure 6 in Appendix C.1).

### 4.2. Curling Dynamics: Ocean vortex data

We test WLM's ability to learn the curling dynamics of particles within an ocean vortex, whose velocity field was measured in the Gulf of Mexico. We consider the interpolation experiment from previous works (Shen et al., 2024; Petrović et al., 2025): given that 111 particles spiral out from the eye of the vortex, and are observed at 5 uniformly spaced times (train on $t_1, t_3, t_5, t_7, t_9$), the task is to interpolate 4 intermediary marginals (test on $t_2, t_4, t_6, t_8$). We also consider the same forecasting experiment from Berlinghieri et al. (2025), which considers a larger vortex from the same dataset. There, the task is to predict the last marginal, after training on 10 previous marginals of 400 samples each. For the interpolation task, we add time features to model a time-varying potential energy $\mathcal{U}[\rho_t, t]$. To perform the forecasting task, we do not consider time features. For more implementation details, see Appendix C.2.

We report results for interpolation and forecasting in Table 2, noting that only WLM, SBIRR, and SNAP-MMD are capable of forecasting. WLM achieves the best interpolations among methods that do not use a reference process, and the second best overall performance after CURLY-FM, which trains a flow matching method based on a 'curly' reference process constructed from velocity fields at all training marginals. In contrast, WLM learns a mechanics model directly from the observed marginals and the initial velocity. We visualize WLM's ability to predict unseen marginals along the vortex's curvature in Figure 4a. Conversely, gradient-flow and OT-based methods struggle to interpolate along the vortex, with the notable exception of OT-MFM, which leverages the data manifold structure. We also emphasize that by learning the mechanics, WLM is able to forecast, whereas flow matching methods like CURLY-FM can only interpolate. While SBIRR (Shen et al., 2024) and SNAP-MMD (Berlinghieri et al., 2025) produce reasonable forecasts by using reference SDEs based on vortex dynamics (Shen et al., 2024, D.5.2) (Berlinghieri et al., 2025, D.9.2), WLM produces more accurate forecasts by directly learning second-order mechanics.

### 4.3. Cell Dynamics: Embryonic scRNA Data

Inferring the developmental trajectories of evolving cells is a notoriously challenging problem, due to its biological complexity, as well as inherent challenges in the sc-RNA measurement process, which is necessarily destructive (Trapnell et al., 2014). To test WLM's ability to learn the evolution of cell populations, we consider the leave-one-out interpolation task for each of the three intermediate time marginals of the embryoid body (EB) dataset Moon et al. (2019). The EB dataset measures the differentiation of human embryonic stem cells over the course of 27 days, such that the data is split into 5 marginals of sizes 2381, 4163, 3278, 3665 and 3332. By following the standard preprocessing of whitening

*Table 2.* $W_1$ distance (averaged over 3 seeds) for interpolating and forecasting on the Gulf of Mexico dataset. We underline best results for methods that use a reference process from domain knowledge and bold best results for methods that do not.

| Method | Interpolation (small vortex) | | | | Forecast (big vortex) | |
|---|---|---|---|---|---|---|
| | $t_2$ | $t_4$ | $t_6$ | $t_8$ | $t_{11}$ (from $t_0$) | $t_{11}$ (from $t_{10}$) |
| **Uses reference process from domain knowledge** | | | | | | |
| CURLY-FM* (Petrović et al., 2025) | 0.019±0.003 | 0.045±0.005 | 0.027±0.001 | 0.030±0.006 | – | – |
| SBIRR (Shen et al., 2024) | 0.073±0.035 | 0.087±0.033 | 0.062±0.010 | 0.082±0.022 | 1.062±0.111 | 0.567±0.014 |
| SNAP-MMD (Berlinghieri et al., 2025) | 0.051±0.003 | 0.067±0.002 | 0.131±0.005 | 0.056±0.001 | 0.896±0.141 | 0.473±0.013 |
| **No reference process from domain knowledge** | | | | | | |
| AM (Neklyudov et al., 2023a) | 0.358±0.009 | 0.353±0.020 | 0.447±0.013 | 0.346±0.016 | – | – |
| UAM (Neklyudov et al., 2023a) | 0.291±0.008 | 0.378±0.021 | 0.514±0.010 | 0.349±0.009 | – | – |
| SAM (Neklyudov et al., 2023a) | 0.358±0.009 | 0.354±0.010 | 0.451±0.016 | 0.350±0.009 | – | – |
| DICE (Blickhan et al., 2025) | 0.369±0.019 | 0.252±0.020 | 0.240±0.011 | 0.115±0.009 | – | – |
| VANILLA-SB (Shen et al., 2024) | 0.112±0.061 | 0.178±0.073 | 0.262±0.043 | 0.305±0.032 | – | – |
| OT-CFM* (Tong et al., 2023a) | 0.148±0.004 | 0.227±0.008 | 0.191±0.012 | 0.250±0.018 | – | – |
| OT-MFM* (Kapusniak et al., 2024) | 0.107±0.014 | 0.056±0.014 | **0.052±0.011** | 0.070±0.021 | – | – |
| WLM (Ours) | **0.039±0.002** | **0.028±0.000** | 0.083±0.002 | **0.050±0.004** | **0.180±0.038** | **0.065±0.005** |

*Numbers taken from Petrović et al. (2025)

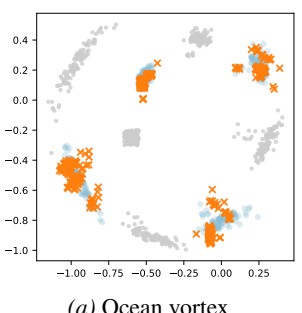
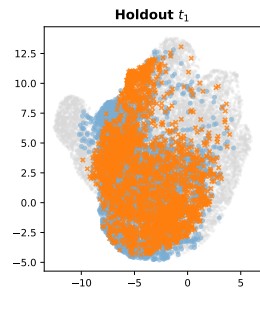
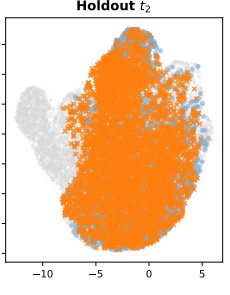
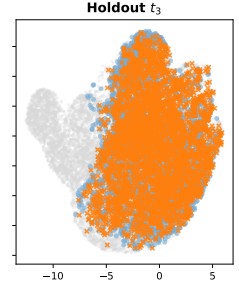

*(a)* Ocean vortex        *(b)* Embryoid body

*Figure 4.* WLM's predictions (×) for unseen interpolants (●) are visualized for the (a) Ocean vortex (in spatial coordinates) and (b) Embryoid body (in UMAP coordinates) datasets. Marginals that are used to train WLM's mechanics model are plotted in gray.

*Table 3.* Average $W_1$ distance for leave-one-out interpolation on 5-dim PCA representation of EB scRNA dataset. Results are averaged over 5 runs.

| Method | $W_1$ |
|---|---|
| REG. CNF* (Finlay et al., 2020) | 0.825 ± 0.429 |
| TRAJECTORYNET* (Tong et al., 2020) | 0.848 |
| NLSB* (Koshizuka & Sato, 2022) | 0.970 |
| DSBM* (Shi et al., 2024) | 1.775 ± 0.429 |
| DSB* (De Bortoli et al., 2021) | 0.862 ± 0.023 |
| SF²M-SINK* (Tong et al., 2023b) | 1.198 ± 0.342 |
| SF²M-GEO* (Tong et al., 2023b) | 0.879 ± 0.148 |
| SF²M-EXACT* (Tong et al., 2023b) | 0.793 ± 0.066 |
| OT-CFM* (Tong et al., 2023a) | 0.790 ± 0.068 |
| I-CFM* (Tong et al., 2023a) | 0.872 ± 0.087 |
| SB-CFM* (Tong et al., 2023a) | 1.221 ± 0.380 |
| WLF-UOT* (Neklyudov et al., 2023b) | 0.738 ± 0.014 |
| WLF-SB* (Neklyudov et al., 2023b) | 0.746 ± 0.016 |
| WLF-OT* (Neklyudov et al., 2023b) | 0.742 ± 0.012 |
| I-MFM* (Kapusniak et al., 2024) | 0.822 ± 0.042 |
| OT-MFM* (Kapusniak et al., 2024) | 0.713 ± 0.039 |
| AM (Neklyudov et al., 2023a) | 0.944 ± 0.003 |
| UAM (Neklyudov et al., 2023a) | 0.924 ± 0.004 |
| SAM (Neklyudov et al., 2023a) | 0.939 ± 0.003 |
| DICE (Blickhan et al., 2025) | 0.826 ± 0.002 |
| WLM (Ours) | **0.704 ± 0.021** |

*Numbers taken from Kapusniak et al. (2024)

the 5-dimensional PCA data (Tong et al., 2020), we compare WLM against a comprehensive list of methods evaluated on the same experiment (Kapusniak et al., 2024).

We implement WLM without including time features in the potential energy $\mathcal{U}[\rho_t]$ and report the results in Table 3. Even without leveraging explicit temporal information, WLM outperforms existing state-of-the-art methods. We summarize a few key insights. First, since cell dynamics remain unknown, a common approach has been to guess different priors and hope to produce developmental trajectories that have a reasonable curvature. Indeed, many of the methods in Table 3 use different optimal transport or Lagrangian priors, which can significantly influence inference quality. In contrast, WLM directly learns population mechanics from the data, without specifying a reference process. Furthermore, WLM's learned time-independent potential energy and friction can be interpreted in order to gain insight into the nature of the dynamics. In this experiment, we recover dissipative dynamics from moderate to high friction (see Figure 7), which is consistent with the fact that mathematical biologists commonly use gradient flows to model cell dynamics (Waddington, 2015; Weinreb et al., 2018; Lavenant et al., 2021). Finally, WLM's result on this dataset can likely be improved, since we did not use a validation set for early-stopping, as done in OT-MFM (Kapusniak et al., 2024). We report an additional ablation to investigate WLM's performance ($W_1$ on holdouts) and runtime when populations

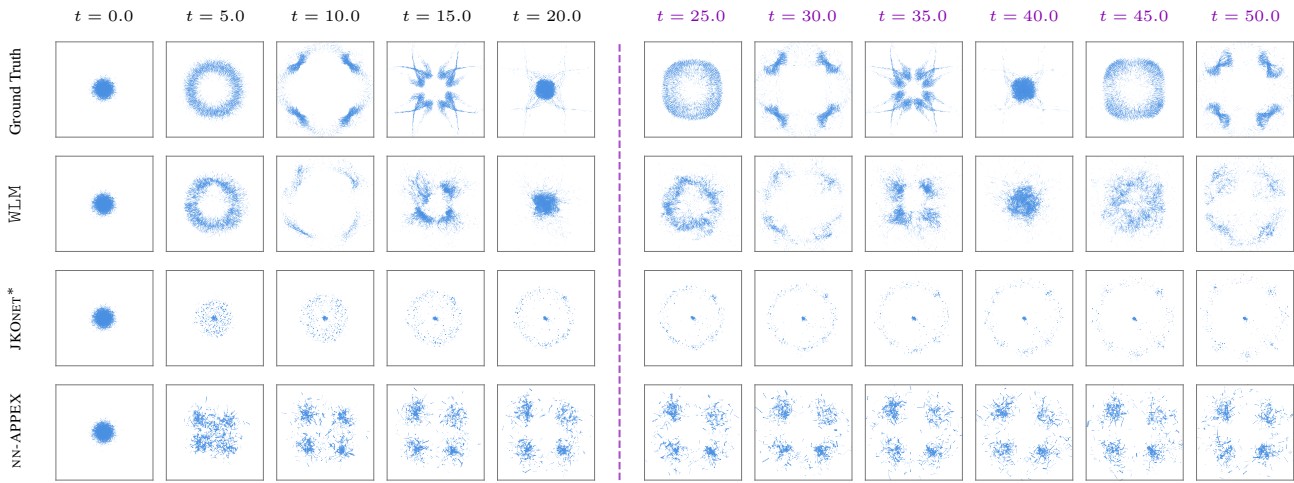

*Figure 5.* **Learning Boids:** We plot the ground truth Boids (first row) against the population dynamics learned by our method `WLM` (second row), and the gradient flow baselines JKONET* (third row) and NN-APPEX (fourth row). All methods were trained on 50 marginals of size 1000 within $t \in [0, 24.5]$. The rollouts of each method are simulated from time 0 (with 5000 samples drawn) until the additional forecast horizon $[25.0, 50.0]$ (indicated by purple time labels). We include additional figures of how the model learned by `WLM` generalizes to Boids dynamics from unseen initial populations in Figure 9.

*Table 4.* Boids: Average $W_1$ distances between true and learned marginals. We distinguish between train times (first 50) and forecast (next 50).

| Method | Train $W_1$ ($\pm$SE) | Forecast $W_1$ ($\pm$SE) |
|---|---|---|
| JKONET* | $3.084 \pm 0.229$ | $3.174 \pm 0.084$ |
| JKONET* (time-var.) | $4.400 \pm 0.229$ | $10.235 \pm 0.341$ |
| NN-APPEX | $2.467 \pm 0.193$ | $3.078 \pm 0.199$ |
| WLM (ours) | $\mathbf{0.496 \pm 0.024}$ | $\mathbf{1.309 \pm 0.034}$ |

are subsampled as mini-batches during training. We report results in Figure 8 and note that in addition to decreasing runtime, the best performance results for this experiment were obtained with mini-batching ($n = 1024$).

### 4.4. Emergent dynamics: Boids

Murmuration is a swarming behaviour observed in large flocks of birds, whose collective flight produces intricate patterns in the sky. Boids is an influential model that was developed by Reynolds (1987). Indeed, Boids has been used to simulate swarms in high-budget movies [1] (Bajec & Heppner, 2009) and is also used to model cognition (Rosas et al., 2020; Lawrence & Nehaniv, 2024) and automated vehicles (Saska et al., 2014; Knievel et al., 2023). The algorithm simulates interacting agents that obey three rules:

1. ***Separation***: steer to avoid colliding with flockmates,
2. ***Alignment***: steer towards average heading of flock,
3. ***Cohesion***: steer towards the flock's centre of mass.

While there are many approaches for simulating Boids (Schoenholz & Cubuk, 2020; Hartman & Benes, 2006), analytic solutions are intractable due to the complexity of

the emergent dynamics. Given the absence of a reference process, we consider the gradient flow methods JKONET* and NN-APPEX as baselines. To model mechanics that are completely determined by the population's potential energy, we implement `WLM` with $\gamma = 0$ and do not model time-dependence. All methods are trained on 50 evenly spaced marginals with 1000 Boids per time. We then reserve the next 50 marginals as the test set to evaluate the ability of all methods to forecast the dynamics.

We present numerical results in Table 4 and qualitative results in Figure 5. While JKONET* and NN-APPEX cannot even fit the observed dynamics, as they only simulate slowly diffusing agents, `WLM` learns population mechanics that cohere to the Boids' cohesion and repulsion agent-based rules. Indeed, `WLM`'s simulated flocks match each of the true Boids' outward and inward migrations, even when forecasting 50 marginals beyond its training data. Furthermore, we observe generalization across unseen initial conditions. Distinct Boids dynamics emerge from different initial populations, and `WLM` can predict these dynamics on unseen populations. We visualize this in Figure 9, and provide interactive examples in our notebook [2]. Overall, these results show that gradient flows generally fail to capture rule-based agentic systems, e.g. due to periodicity. Meanwhile, `WLM`'s ability to fit the Boids dynamics, and generalize to unseen data (beyond the training horizon and across initial conditions), suggests that Boids may admit a variational principle.

As an additional experiment, we test `WLM`'s robustness for learning Boids dynamics when the initial velocity is mis-

---

[1]e.g. bat swarms and penguin flocks in Batman Returns (1992)

[2]https://colab.research.google.com/drive/
1R4q5pysHaYOMIu77vT6JID7yCLYG3t4z?usp=sharing

specified. We either completely misspecify the initial velocity as 0, or we estimate it from the data, using DICE (Blickhan et al., 2025). While access to the true initial velocity helps the model, accurate forecasts are still achieved when the velocity is misspecified (see Table 7).

## 5. Related Work

**Wasserstein Lagrangian and Hamiltonian Flows** The mathematical literature has developed Lagrangian and Hamiltonian formulations for modeling conservative dynamics in the Wasserstein space (Ambrosio & Gangbo, 2008; Chow et al., 2020). Neklyudov et al. (2023b) was the first work to use Wasserstein Lagrangian flows as a model for inferring population dynamics given observational data. The main practical limitation of this work and related methods (Qian et al., 2024; Sun et al., 2025) is that the Lagrangian formulation requires a reference process in order to obtain a well-defined action minimization (3) for inferring interpolants. In our work, we derive the Hamiltonian mechanics from *damped Wasserstein Lagrangian flows*. This allows our method, WLM, to interpolate and forecast, without prespecifying the Lagrangian. The incorporation of damping also allows WLM to learn dissipative population dynamics, which provide a stable representation of Wasserstein gradient flows in the overdamped limit.

**Wasserstein Gradient Flows.** Gradient flows are the predominant model for population dynamics (Hashimoto et al., 2016; Weinreb et al., 2018; Lavenant et al., 2021; Bunne et al., 2022; Neklyudov et al., 2023a; Terpin et al., 2024; Guan et al., 2026). In this work, we build on mathematical ideas from Wasserstein gradient flows (Jordan et al., 1998; Ambrosio et al., 2008) in order to develop WLM as a richer class of second-order dynamics on the same space. We then demonstrate that WLM learns gradient flows as effectively as state-of-the-art gradient flow methods, while also effectively learning non-gradient dynamics, like Boids.

**Reference-based Population Dynamics.** Many methods infer population dynamics given a reference process (Schiebinger et al., 2019; Bunne et al., 2023; Berlinghieri et al., 2025; Petrović et al., 2025), or family of reference processes (Shen et al., 2024; Zhang, 2024; Guan et al., 2024; 2026). Reference-based inference is often done to overcome the limitations of gradient flows (Shen et al., 2024; Zhang, 2024; Guan et al., 2024; Petrović et al., 2025; Berlinghieri et al., 2025). Our work also addresses the limitations of gradient flows, but WLM is not reference-based, and is therefore less susceptible to model misspecification.

**Flow and Action Matching.** Flow matching (Lipman et al., 2022) and action matching (Neklyudov et al., 2023a) have been proposed to learn continuous interpolations from a finite set of time marginals, and have each inspired methods tailored to scientific inference (Atanackovic et al., 2024;

Kapusniak et al., 2024; Blickhan et al., 2025). However, unlike WLM, these methods cannot forecast, since they are only trained to interpolate within the observation period.

## 6. Conclusion

In this work, we proposed and advanced a new direction for inferring population dynamics, by emphasizing the *population mechanics* as an intrinsic and learnable property. We developed a comprehensive theory for *Wasserstein Lagrangian Mechanics* by considering a population-level principle of least action with damping, which produces second-order Hamiltonian mechanics that encompass classical mechanics, quantum mechanics, and gradient flows. We then leverage these insights to develop WLM, a powerful inference method that outperforms state-of-the-art methods for interpolating and forecasting unseen marginals on real and synthetic datasets from the literature. WLM also learns the emergent dynamics from interacting Boids, demonstrating exciting new capabilities from learning the mechanics.

**Limitations and Directions for Future Work.** While we believe that this work makes important theoretical and practical advances, the learning of population mechanics is a nascent field. We outline limitations from our paper, which may provide interesting opportunities for future work.

There are many open questions about the expressivity and identifiability of WLM systems. While WLM can describe classical mechanics, quantum mechanics, and gradient flows, the population dynamics of other systems may not obey a Wasserstein least action principle. Given WLM's ability to learn Boids, it would be interesting to determine conditions such that a least action principle holds for interacting agents. With respect to identifiability, we have already seen that gradient flows admit multiple WLM formulations in Proposition 2.4. We conjecture that under certain assumptions, like $\gamma = 0$, non-degeneracy, and non-equilibrium, a wide class of WLM systems can be uniquely identified from continuous population dynamics. Furthermore, without imposing additional assumptions, various individual trajectories produce the same population dynamics. In this work, we model trajectories using the ODE from the canonical continuity equation. Interesting future work remains for modeling other trajectories of interest, such as solutions to the Navier-Stokes equations or solutions to the Schrödinger equation.

For inference, the largest practical barrier is that WLM simulates dynamics to train the mechanics model. Given sufficiently dense data, a simulation free approach may be feasible by first learning the vector field $\nabla s_t$, and then extracting the Hamiltonian mechanics. Our initial attempts were unsuccessful, as any errors incurred by the estimation are inherited downstream.

## Impact Statement

This work is primarily a theoretical and methodological contribution towards scientific inference, and poses little risk for social harm. Better prediction of population dynamics could potentially be used for malicious development, but we do not see this as a significant risk at this time, as the theory primarily pertains to cellular and molecular systems.

## Acknowledgments

VG was supported by the PGS-D scholarship, funded by the Natural Sciences and Engineering Research Council of Canada, and was affiliated with Mila as a visiting researcher. The research was enabled in part by computational resources provided by the Digital Research Alliance of Canada (https://alliancecan.ca) and Mila (https://mila.quebec). In addition, KN was supported by IVADO and Institut Courtois. LA was in-part supported by the Eric and Wendy Schmidt Center at the Broad Institute of MIT and Harvard, and by the NSERC Postdoctoral Fellowship. This project was undertaken thanks to funding from IVADO and the Canada First Research Excellence Fund.

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

## A. Additional Proofs

**Theorem A.1** (Hamiltonian equations of motion). *Let $(p_t)_{0 \leq t \leq 1}$ minimize action with respect to the damped Wasserstein Lagrangian*

$$\mathcal{L}[\rho_t, s_t, t] = e^{\gamma t} \left( \frac{1}{2} \int \|\nabla s_t\|^2 \rho_t dx - \mathcal{U}[\rho_t] \right) \tag{11}$$

*with damping $\gamma \geq 0$. Then, $(p_t)_{0 \leq t \leq 1}$ obeys the continuity equation (1), such that the driving potential $s_t$ obeys*

$$\dot{s}_t(x) = -\frac{1}{2} \|\nabla s_t(x)\|^2 - \frac{\delta \mathcal{U}[\rho_t]}{\delta \rho_t}(x) - \gamma s_t(x). \tag{12}$$

*The marginals $(p_t)_{0 \leq t \leq 1}$ can therefore be produced by sampling trajectories, obeying the second order mechanics:*

$$\frac{d}{dt} x_t = v_t := \nabla s_t(x_t), \qquad\qquad x_0 \sim p_0 \tag{13}$$

$$\frac{d}{dt} v_t = -\nabla \frac{\delta \mathcal{U}[\rho_t]}{\delta \rho_t}(x_t) - \gamma v_t, \qquad\qquad v_0 = \nabla s_0(x_0) \tag{14}$$

*Proof.* The action functional is given by

$$\mathcal{S}[\rho_t, s_t] = \int_0^1 e^{\gamma t} \left( \frac{1}{2} \int \|\nabla s_t\|^2 \rho_t dx - \mathcal{U}[\rho_t] \right) dt. \tag{15}$$

First, we note that by the continuity equation (1), the term $\int_0^1 e^{\gamma t} \int s_t \left( \dot{\rho}_t + \nabla \cdot (\rho_t \nabla s_t) \right) dx dt$ is identically 0, so we can add it to the damped action functional (15). Integrating by parts allows us to rewrite $\mathcal{S}[\rho_t, s_t]$ as

$$\mathcal{S}[\rho_t, s_t] = \int_0^1 e^{\gamma t} \left( -\frac{1}{2} \int \|\nabla s_t\|^2 \rho_t dx - \mathcal{U}[\rho_t] + \int s_t \dot{\rho}_t dx \right) dt.$$

Then, by the principle of least action, we have the optimality conditions (4). While $\frac{\delta \mathcal{S}[\rho_t, s_t]}{\delta s_t}[h_t] = 0$ simply recovers the continuity equation (1), we can use $\frac{\delta \mathcal{S}[\rho_t, s_t]}{\delta \rho_t}[h_t] = 0$ to derive the evolution of $s_t$. Integrating by parts yields

$$0 = \int_0^1 e^{\gamma t} \int \left( -\frac{1}{2} h_t \|\nabla s_t\|^2 - h_t \frac{\delta \mathcal{U}[\rho_t]}{\delta \rho_t} + s_t \dot{h}_t \right) dx dt$$

$$= \int_0^1 e^{\gamma t} \int h_t \underbrace{\left( -\frac{1}{2} \|\nabla s_t\|^2 - \frac{\delta \mathcal{U}[\rho_t]}{\delta \rho_t} - \dot{s}_t - \gamma s_t \right)}_{=0} dx dt$$

We therefore obtain the evolution equation of the driving potential (6), since $e^{\gamma t} > 0$ and the above holds for all test functions $h_t$. Then, to determine the induced trajectory $(x_t)_{0 \leq t \leq 1}$ for a particle $x_0 \sim p_0$, we set $v_t(x) = \nabla s_t(x)$ and compute the total derivative of the particle's time-varying velocity, $\dot{x}_t = v_t(x_t)$:

$$\frac{d}{dt} v_t(x_t) = \left( \partial_t v_t + (v_t \cdot \nabla) v_t \right)(x_t), \tag{16}$$

where we clarify that the operator $v_t \cdot \nabla = \sum_{i=1}^d v_t^{(i)} \frac{\partial}{\partial x^{(i)}}$ and hence, $v_t \cdot \nabla v_t$ is a vector with components

$$[v_t \cdot \nabla v]_j = \sum_{i=1}^d v_t^{(i)} \frac{\partial v_t^{(j)}}{\partial x^{(i)}}$$

The first term in (16) is computed by taking the gradient on both sides of the evolution equation of $s_t$ (6):

$$\partial_t v_t = -\nabla \left( \frac{\delta \mathcal{U}[\rho_t]}{\delta \rho_t} \right) - \gamma v_t - \nabla (\frac{1}{2} \|v_t\|^2).$$

Then, the second term simplifies to $(v_t \cdot \nabla)v_t = \nabla(\frac{1}{2}\|v_t\|^2)$, since $v_t = \nabla s_t$ is conservative. It follows that

$$\frac{d}{dt}v_t = -\nabla \frac{\delta \mathcal{U}[\rho_t]}{\delta \rho_t}(x_t) - \gamma v_t, \quad v_0 = \nabla s_0(x_0)$$

$\square$

Intuitively, the potential energy determines the evolution of the velocity field, which determines the population dynamics. These population dynamics are attained by aggregating paths from the corresponding ODE.

**Proposition A.2** (Wasserstein Lagrangian of a gradient flow). *Let $\gamma = 0$. Then, $(\rho_t)_{0 \leq t \leq 1}$ is a gradient flow of $\mathcal{F}[\rho_t]$ if and only if it minimizes the Wasserstein Lagrangian action*

$$\mathcal{S}[\rho_t, s_t] = \int_0^1 \{\mathcal{K}[\rho_t, s_t] - \mathcal{U}[\rho_t]\}\, dt \tag{17}$$

$$:= \int_0^1 \left\{\frac{1}{2}\int \|\nabla s_t\|^2 \rho_t\, dx + \frac{1}{2}\int \|\nabla \frac{\delta}{\delta \rho_t}\mathcal{F}[\rho_t]\|^2 \rho_t\, dx\right\} dt, \tag{18}$$

*and additionally has initial velocity $v_0 \propto -\nabla \frac{\delta}{\delta \rho_0}\mathcal{F}[\rho_0]$.*

*Proof.* Recall that population dynamics $(\rho_t)_{0 \leq t \leq 1}$ are a Wasserstein gradient flow of $\mathcal{F}$ if and only if $\nabla s_t \propto -\nabla \frac{\delta}{\delta \rho_t}\mathcal{F}(\rho_t)$ for $\rho_t$ a.e. $(x, t) \in \mathbb{R}^d \times [0, 1]$ (Santambrogio, 2017). We first prove that a necessary condition is that the dynamics minimize the Wasserstein Lagrangian action given by the $W_2$ kinetic energy (2) and the potential energy $\mathcal{U}[\rho_t] = -\frac{1}{2}\int \|\nabla \frac{\delta}{\delta \rho_t}\mathcal{F}[\rho_t]\|^2 \rho_t dx$.

Let $\mathcal{F}$ be a sufficiently regular free energy functional so that $\frac{\delta}{\delta \rho_t}\mathcal{F}(\rho_t)$ exists and let $\rho_0$ and $\rho_1$ be two observed marginals. By definition, if $(\rho_t)_{0 \leq t \leq 1}$ is a gradient flow for $\mathcal{F}$, then it minimizes the non-negative objective

$$\mathcal{J}[\rho_t, s_t] = \frac{1}{2}\int_0^1 \int \|\nabla s_t + \nabla \frac{\delta}{\delta \rho_t}\mathcal{F}[\rho_t]\|^2 \rho_t\, dx\, dt, \tag{19}$$

Thus, given fixed $\rho_0 = p_0$ and $\rho_1 = p_1$, we seek to minimize (19). However, we may rewrite

$$\mathcal{J}[\rho_t, s_t] = \int_0^1 \left\{\frac{1}{2}\int \|\nabla s_t\|^2 \rho_t\, dx + \frac{1}{2}\int \|\nabla \frac{\delta}{\delta \rho_t}\mathcal{F}[\rho_t]\|^2 \rho_t\, dx\right\} dt + \int_0^1 \int \nabla \frac{\delta}{\delta \rho_t}\mathcal{F}[\rho_t] \cdot \nabla s_t \rho_t\, dx\, dt$$

$$= \int_0^1 \left\{\frac{1}{2}\int \|\nabla s_t\|^2 \rho_t\, dx + \frac{1}{2}\int \|\nabla \frac{\delta}{\delta \rho_t}\mathcal{F}[\rho_t]\|^2 \rho_t\, dx\right\} dt - \int_0^1 \int \frac{\delta}{\delta \rho_t}\mathcal{F}[\rho_t]\nabla \cdot (\nabla s_t \rho_t)\, dx\, dt$$

$$= \int_0^1 \left\{\frac{1}{2}\int \|\nabla s_t\|^2 \rho_t\, dx + \frac{1}{2}\int \|\nabla \frac{\delta}{\delta \rho_t}\mathcal{F}[\rho_t]\|^2 \rho_t\, dx\right\} dt + \int_0^1 \int \frac{\delta}{\delta \rho_t}\mathcal{F}[\rho_t]\frac{\partial}{\partial t}\rho_t\, dx\, dt$$

$$= \int_0^1 \left\{\frac{1}{2}\int \|\nabla s_t\|^2 \rho_t\, dx + \frac{1}{2}\int \|\nabla \frac{\delta}{\delta \rho_t}\mathcal{F}[\rho_t]\|^2 \rho_t\, dx\right\} dt + \int_0^1 \frac{d}{dt}\mathcal{F}[\rho_t]\, dt$$

$$= \int_0^1 \left\{\frac{1}{2}\int \|\nabla s_t\|^2 \rho_t\, dx + \frac{1}{2}\int \|\nabla \frac{\delta}{\delta \rho_t}\mathcal{F}[\rho_t]\|^2 \rho_t\, dx\right\} dt + \mathcal{F}[\rho_1] - \mathcal{F}[\rho_0],$$

where we expanded the square, integrated by parts, applied the continuity equation, and the chain rule. However, $\mathcal{F}[\rho_1]$ and $\mathcal{F}[\rho_0]$ are independent of the minimization, since the endpoints are fixed. Thus, minimizing $\mathcal{J}[\rho_t, s_t]$ is equivalent to minimizing

$$\mathcal{S}[\rho_t, s_t] = \int_0^1 \{\mathcal{K}[\rho_t, s_t] - \mathcal{U}[\rho_t]\}\, dt \tag{20}$$

$$:= \int_0^1 \left\{\frac{1}{2}\int \|\nabla s_t\|^2 \rho_t\, dx + \frac{1}{2}\int \|\nabla \frac{\delta}{\delta \rho_t}\mathcal{F}[\rho_t]\|^2 \rho_t\, dx\right\} dt, \tag{21}$$

which is precisely the Wasserstein Lagrangian action with canonical kinetic energy $\mathcal{K}[\rho_t, s_t]$(2) and potential energy $\mathcal{U}[\rho_t] = -\frac{1}{2} \int \|\nabla \frac{\delta}{\delta \rho_t} \mathcal{F}[\rho_t]\|^2 \rho_t dx$. Thus, action minimization with respect to these Wasserstein energies is a necessary condition to be a gradient flow.

Now, given that the marginals $(\rho_t)_{0 \leq t \leq 1}$ minimize the action $\mathcal{S}$, we want to show that $(\rho_t)_{0 \leq t \leq 1}$ is a gradient flow of $\mathcal{F}$ if and only if we have the precise initial velocity $v_0 \propto -\nabla \frac{\delta \mathcal{F}[\rho_0]}{\delta \rho_0}$. Since a gradient flow must have $\nabla s_t(x) \propto -\nabla \frac{\delta}{\delta \rho_t} \mathcal{F}(\rho_t)(x)$ almost everywhere, the idea is to define a discrepancy function to clarify the analysis. Let $\lambda$ be the scalar proportion and

$$g_t(x) = v_t(x) + \lambda \nabla \frac{\delta \mathcal{F}[\rho_t]}{\delta \rho_t}(x). \tag{22}$$

By definition, the dynamics are a gradient flow if and only if $g_t = 0$ for all $t \in [0, 1]$. We would then complete the proof of the proposition by showing that $\frac{d}{dt} g_t(x_t)$ defines a homogeneous linear evolution, since this would imply that the only valid solution is $g_0 = 0 \implies v_0 \propto -\nabla \frac{\delta \mathcal{F}[\rho_0]}{\delta \rho_0}$. The rest of the proof consists in computations to demonstrate this. We set $\lambda = 1$ and note that the steps are equivalent for other values of $\lambda$. First, we compute

$$\frac{d}{dt} g_t = \frac{d}{dt} v_t + \frac{d}{dt} \nabla \frac{\delta \mathcal{F}[\rho_t]}{\delta \rho_t}(x_t) = \frac{d}{dt} v_t + \nabla \frac{\partial}{\partial t} \frac{\delta \mathcal{F}[\rho_t]}{\delta \rho_t} + \nabla^2 \frac{\delta \mathcal{F}[\rho_t]}{\delta \rho_t} \cdot v_t. \tag{23}$$

We compute the first term by considering the Hamiltonian equation $\frac{d}{dt} v_t = -\nabla \frac{\delta \mathcal{U}}{\delta \rho_t}$ (8) with the potential energy that we previously derived, $\mathcal{U}[\rho_t] = -\frac{1}{2} \int \|\nabla \frac{\delta \mathcal{F}[\rho_t]}{\delta \rho_t}\|^2 \rho_t \, dx$. We have that

$$\frac{\delta \mathcal{U}}{\delta \rho} = -\frac{1}{2} \|\nabla \frac{\delta \mathcal{F}}{\delta \rho}\|^2 + \int \frac{\delta^2 \mathcal{F}}{\delta \rho(x) \delta \rho(y)} \nabla \cdot (\rho_t(y) \nabla \frac{\delta \mathcal{F}}{\delta \rho_t}(y)) dy$$

it follows that

$$\frac{d}{dt} v_t = \nabla \left( \frac{1}{2} \left\| \nabla \frac{\delta \mathcal{F}[\rho_t]}{\delta \rho_t} \right\|^2 \right) - \nabla \int \frac{\delta^2 \mathcal{F}}{\delta \rho_t(x) \delta \rho_t(y)} \nabla \cdot \left( \rho_t(y) \nabla \frac{\delta \mathcal{F}[\rho_t]}{\delta \rho_t}(y) \right) dy. \tag{24}$$

Then, using the continuity equation $\dot{\rho}_t = -\nabla \cdot (\rho_t v_t)$ and chain rule, the second term is

$$\nabla \frac{\partial}{\partial t} \frac{\delta \mathcal{F}[\rho_t]}{\delta \rho_t} = \nabla \int \frac{\delta^2 \mathcal{F}}{\delta \rho_t(x) \delta \rho_t(y)} \dot{\rho}_t(y) dy = -\nabla \int \frac{\delta^2 \mathcal{F}}{\delta \rho_t(x) \delta \rho_t(y)} \nabla \cdot (\rho_t(y) v_t(y)) dy. \tag{25}$$

We plug these terms back into the expression for $\frac{d}{dt} g_t$ (23), separating all of the integral terms from the non-integral terms:

$$\frac{d}{dt} g_t = A - B \tag{26}$$

$$A = \left[ \nabla \left( \frac{1}{2} \left\| \nabla \frac{\delta \mathcal{F}[\rho_t]}{\delta \rho_t} \right\|^2 \right) + \nabla^2 \frac{\delta \mathcal{F}[\rho_t]}{\delta \rho_t} \cdot v_t \right] \tag{27}$$

$$B = \left[ \nabla \int \frac{\delta^2 \mathcal{F}}{\delta \rho_t(x) \delta \rho_t(y)} \nabla \cdot \left( \rho_t(y) \nabla \frac{\delta \mathcal{F}[\rho_t]}{\delta \rho_t}(y) \right) dy + \nabla \int \frac{\delta^2 \mathcal{F}}{\delta \rho_t(x) \delta \rho_t(y)} \nabla \cdot (\rho_t(y) v_t(y)) dy \right] \tag{28}$$

We simplify the first term in $A$ with the vector identity $\nabla(\frac{1}{2} \|\nabla f\|^2) = (\nabla^2 f) \cdot \nabla f$ to factor out $g_t$ as desired:

$$A = \left( \nabla^2 \frac{\delta \mathcal{F}}{\delta \rho_t} \right) \cdot \nabla \frac{\delta \mathcal{F}}{\delta \rho_t} + \left( \nabla^2 \frac{\delta \mathcal{F}}{\delta \rho_t} \right) \cdot v_t \tag{29}$$

$$= \left( \nabla^2 \frac{\delta \mathcal{F}}{\delta \rho_t} \right) \cdot \left( \nabla \frac{\delta \mathcal{F}}{\delta \rho_t} + v_t \right) = \left( \nabla^2 \frac{\delta \mathcal{F}}{\delta \rho_t} \right) \cdot g_t \tag{30}$$

We do a similar process for the integral terms in $B$, just by using the linearity of the divergence operator:

$$B = \nabla \int \frac{\delta^2 \mathcal{F}}{\delta \rho_t(x) \delta \rho_t(y)} \left[ \nabla \cdot \left( \rho_t \nabla \frac{\delta \mathcal{F}}{\delta \rho_t} \right) + \nabla \cdot (\rho_t v_t) \right] dy \tag{31}$$

$$= \nabla \int \frac{\delta^2 \mathcal{F}}{\delta \rho_t(x) \delta \rho_t(y)} \nabla \cdot \left( \rho_t(y) \left[ v_t(y) + \nabla \frac{\delta \mathcal{F}}{\delta \rho_t}(y) \right] \right) dy \tag{32}$$

$$= \nabla \int \frac{\delta^2 \mathcal{F}}{\delta \rho_t(x) \delta \rho_t(y)} \nabla \cdot (\rho_t(y) g_t(y)) dy \tag{33}$$

Thus, $\frac{d}{dt} g_t = A - B$ is homogeneous and linear in $g_t$, which admits 0 as solution. By uniqueness under standard regularity, we have that $g_t = 0$ for all $t \geq 0$ as desired. $\qquad\square$

*Proposition* 3.1. Consider the empirical measure $\hat{p} = \frac{1}{N} \sum_{i=1}^{N} \delta_{x^{(i)}}$ and let $\Psi(x^{(1)}, \ldots, x^{(N)}) := \mathcal{U}[\hat{p}]$ be its potential energy. Then, for any particle $x^{(j)} \sim \hat{p}$, we have

$$\nabla_{x^{(j)}} \Psi(x^{(1)}, \ldots, x^{(N)}) = \frac{1}{N} \nabla_x \frac{\delta \mathcal{U}[p]}{\delta p}(x^{(j)}) \Big|_{p=\hat{p}} \tag{9}$$

*Proof.* Since $p = \hat{p} = \frac{1}{N} \sum_{i=1}^{N} \delta_{x^{(i)}}$, it follows that the potential energy supported on the observed samples is equivalent to the true Wasserstein potential energy, i.e., $\Psi(x^{(1)}, \ldots, x^{(N)}) = \mathcal{U}[p]$.

We now prove the identity by considering a perturbation of sample $x^{(j)}$ in the direction of a vector $\eta \in \mathbb{R}^d$, and then matching the corresponding directional derivative for both $\Psi(x^{(1)}, \ldots, x_t^{(N)})$ and $\mathcal{U}[p]$. Indeed, equality implies that their directional derivatives are also the same.

First, we consider $\Psi$ and derive:

$$\frac{d}{d\epsilon} \Big|_{\epsilon=0} \Psi(x^{(1)}, \ldots, x^{(j)} + \epsilon\eta, \ldots, x^{(N)}) = \nabla_{x^{(j)}} \Psi(x^{(1)}, \ldots, x_t^{(N)}) \cdot \eta$$

Then, to evaluate the analogous derivative for $\mathcal{U}[p]$, we first consider the perturbed measure:

$$p_\epsilon = \frac{1}{N} \left( \sum_{i \neq j}^{N} \delta_{x^{(i)}} + \delta_{x^{(j)} + \epsilon\eta} \right) \tag{34}$$

The first order Taylor gives us:

$$\mathcal{U}[p_\epsilon] - \mathcal{U}[p] = \int \frac{\delta}{\delta p} \mathcal{U}[p](x) \delta p(x) dx + o(\epsilon) \tag{35}$$

Plugging in our perturbation then yields:

$$\mathcal{U}[p_\epsilon] - \mathcal{U}[p] = \frac{1}{N} \int \frac{\delta}{\delta p} \mathcal{U}[p](x) (\delta_{x^{(j)} + \epsilon\eta} - \delta_{x^{(j)}}) dx + o(\epsilon) \tag{36}$$

$$= \frac{1}{N} \left( \frac{\delta}{\delta p} \mathcal{U}[p](x^{(j)} + \epsilon\eta) - \frac{\delta}{\delta p} \mathcal{U}[p](x^{(j)}) \right) + o(\epsilon) \tag{37}$$

We then first order Taylor expand $\frac{\delta}{\delta p} \mathcal{U}[p](x^{(j)} + \epsilon\eta)]$ to get (using the chain rule):

$$\mathcal{U}[p_\epsilon] - \mathcal{U}[p] = \frac{1}{N} \left( \epsilon \nabla_x \frac{\delta}{\delta p} \mathcal{U}[p](x^{(j)}) \cdot \eta + o(\epsilon) \right) + o(\epsilon) \tag{38}$$

Hence, it follows that:

$$\frac{d}{d\epsilon} \Big|_{\epsilon=0} \mathcal{U}[p_\epsilon] = \frac{1}{N} \nabla_x \frac{\delta}{\delta p} \mathcal{U}[p](x^{(j)}) \cdot \eta \tag{39}$$

Equating both sides gives us the desired (weak) equation:

$$\frac{d}{d\epsilon} \Big|_{\epsilon=0} \Psi(x^{(1)}, \ldots, x^{(j)} + \epsilon\eta, \ldots, x^{(N)}) = \frac{d}{d\epsilon} \Big|_{\epsilon=0} \mathcal{U}[p_\epsilon]$$

$$\implies \nabla_{x^{(j)}} \Psi(x^{(1)}, \ldots, x_t^{(N)}) \cdot \eta = \frac{1}{N} \nabla_x \frac{\delta}{\delta p} \mathcal{U}[p](x^{(j)}) \cdot \eta$$

$\qquad\square$

## B. Non-identifiability of Individual Trajectories

While Proposition 2.1 shows that the canonical continuity equation (2) can without loss of generality model any continuous evolution of marginals, we note that these marginals can generally be produced by many laws on paths. As a simple example, consider trajectories from a gradient-flow diffusion,

$$\mathrm{d}X_t = -\nabla\Psi(X_t)\mathrm{d}t + \sigma\mathrm{d}W_t. \tag{40}$$

Then, its population dynamics $(p_t)_{0\leq t\leq 1}$ are determined by its Fokker-Planck equation, which can equivalently be expressed via the continuity equation (1) for some corresponding transport vector field $\nabla s_t$. In fact, it is well known that $\nabla s_t$ is determined by

$$\nabla s_t(x) = -\nabla\Psi(x) - \frac{\sigma^2}{2}\nabla\log\rho_t(x) = -\nabla\left(\Psi(x) + \frac{\sigma^2}{2}\log\rho_t(x)\right).$$

Thus, when initialized at $p_0$, the ODE $\frac{d}{dt}x_t = \nabla s_t(x)$ will produce trajectories, which, when aggregated, produce the marginals $(p_t)_{0\leq t\leq 1}$. However, these ODE trajectories will differ compared to the stochastic trajectories from the diffusion process (40). Similarly, non-conservative vector fields can produce the same population dynamics $(p_t)_{0\leq t\leq 1}$.

Additional knowledge is therefore needed to uniquely identify the law on paths from observed population dynamics $(p_t)_{0\leq t\leq 1}$. Indeed, this is the central problem for SDE identifiability (Weinreb et al., 2018; Lavenant et al., 2021; Guan et al., 2024; 2026). Recent work has shown that if we restrict candidate laws to gradient-flow SDEs with drift-diffusion $(-\nabla\Psi, \sigma^2)$, then as long as the observed marginals $(p_t)_{0\leq t\leq 1}$ are not trivially at equilibrium, then the SDE parameters that identify the law on paths is in fact identifiable (Guan et al., 2026). Similar identifiability results hold when the class of SDEs is assumed to be linear and when the marginals obey certain symmetry-breaking conditions (Guan et al., 2024). However, in this work, we do not assume anything about the underlying paths. By only considering the evolution of marginals, we always evolve trajectories according to the canonical vector field in (1) for our theoretical results.

## C. Additional Experiment Details

We present additional details about the datasets, implementation of `WLM` and baseline methods, and analysis of results for each of our four experiments.

**WLM Default Hyperparameters.** In all experiments, we implement the potential energy model $\Psi_\theta : \mathbb{R}^{N\times d} \to \mathbb{R}$ as a neural network equipped with both attention and feed forward layers. We found that the attention mechanism helped the model learn more expressive interacting dynamics. Our default settings used 4 attention heads and 4 stacked blocks. Each block consists of a multi-head self-attention mechanism followed by a feed forward network. The default hidden dimensions is 64 and the default inner dimension from the feed forward network is 512. We used a learning rate of $lr = 1e - 4$ for $\Psi_\theta$ in all experiments. We detail changes from the default hyperparameters in each experiment subsection. We also note that since `WLM` is simulation-based, one important hyperparameter is the number of substeps used between marginals during training. While we always sample 5 substeps at inference time to faithfully simulate from the finalized learned model, we interestingly found that using only 1 substep between marginal often significantly improved the model, so we use this as default. To implement our loss, we by default use the Sinkhorn loss with $p = 2$ from the geomloss package https://www.kernel-operations.io/geomloss/. The kernel blur parameter is estimated from the data. The `WLM` code is implemented in PyTorch.

**Hardware.** All experiments were conducted on an HPC cluster, primarily on NVIDIA GPUs (such as the A100 or RTX 8000 series) on one GPU and $2 - 4$ CPU cores, with between $10 - 50$GB of memory per run. Depending on the size of the dataset, most experiments completed in between $3 - 15$ hours. We present the runtime breakdown per experiment below, noting that time complexity principally depends on the number of particles and training marginals, since `WLM` is trained via simulated rollouts. We explore the effect of mini-batching on runtime and performance for the Embroid Body experiment in Figure 8.

*Table 5.* Runtime per experiment

| Experiment | Particles at $t = 0$ | Number of training marginals | Epochs | Hours |
|---|---|---|---|---|
| GF SDEs | 1000 | 9 | 100,000 | 7.00 ($\pm 0.82$) |
| Boids | 1000 | 49 | 30,000 | 10.88 ($\pm 1.38$) |
| Oceans (small) | 111 | 4 | 50,000 | 1.32 ($\pm 0.04$) |
| Oceans (big) | 400 | 9 | 50,000 | 1.99 ($\pm 0.21$) |
| EB | 1024 (minibatched) | 4 | 10,000 | 0.35 ($\pm 0.09$) |

## C.1. Gradient-flow SDEs

We consider the same five two-dimensional potentials used in the main experiments by Guan et al. (2026):

$$\textbf{Bohachevsky} \qquad V(x) = 10\big(x_1^2 + 2x_2^2 - 0.3\cos(3\pi x_1) - 0.4\cos(4\pi x_2)\big) \tag{41}$$

$$\textbf{Oakley–O'Hagan} \qquad V(x) = 5\sum_{i=1}^{2}\big(\sin(x_i) + \cos(x_i) + x_i^2 + x_i\big) \tag{42}$$

$$\textbf{Quadratic} \qquad V(x) = 5\,\|x\|^2 \tag{43}$$

$$\textbf{Styblinski–Tang} \qquad V(x) = \tfrac{1}{2}\sum_{i=1}^{2}\big(x_i^4 - 16x_i^2 + 5x_i\big) \tag{44}$$

$$\textbf{Wavy plateau} \qquad V(x) = \sum_{i=1}^{2}\big(\cos(\pi x_i) + \tfrac{1}{2}x_i^4 - 3x_i^2 + 1\big) \tag{45}$$

Data is then generated by choosing diffusivity $\sigma^2 = 1$. We sample 10 marginals with equal spacing $\Delta t = 0.01$, initialized from a mean 0 Gaussian, $p_0 \sim \mathcal{N}(0, 0.2)$. We generated the data using the Euler-Maruyama scheme, with 10 substeps per $\Delta t$, i.e. $\Delta t_{EM} = 0.001$. Each of these SDEs converge to their stationary distribution at different rates and thus offers a range of different geometries and dynamics. We note that the Bohachevsky SDE almost immediately converges to its stationary distribution, whereas the Oakley-O'Hagan SDE for example is far from reaching its stationary distribution after 10 steps.

**WLM Implementation.** We use the default hyperparameters and set the number of epochs to $100,000$. The friction learning rate is set to $1e - 2$ when friction is learnable, and set to initial value $\gamma = 1$.

**Implementation of Baselines.** We use default implementations for both JKONET* and NN-APPEX, as done for experiments on the same benchmark dataset. See (Terpin et al., 2024, Appendix C) and (Guan et al., 2026, Section 6) respectively.

## C.2. Ocean Currents

**Dataset Description.** As done in Shen et al. (2024); Berlinghieri et al. (2025); Petrović et al. (2025), we obtain real ocean vortex data from the Gulf of Mexico, from HYbrid Coordinate Ocean Model (HYCOM) https://www.hycom.org/data/gomb0pt01/gom-reanalysis. We follow the same data preprocessing as Shen et al. (2024); Petrović et al. (2025) to reproduce the small vortex, and as done in Berlinghieri et al. (2025) to reproduce the big vortex.

**WLM Implementation.** To implement WLM for interpolation on both vortices, we use one attention head, and for the architecture, we increased the number of hidden dimensions to 256 and reduced the feed forward inner dimension to 64. We set learnable friction in both settings (with learning rate $1e - 3$), but only model time as a feature for interpolation (small vortex), using 16 time features to do so. For both tasks, we train for $50,000$ epochs and use EMA 0.9999. We use weight decay $1e - 2$ for the interpolation task on the small vortex and weight decay $1e - 3$ for the forecast task on the big vortex.

*Table 6.* Comparison of average $W_1$ distances for JKONET*, NN-APPEX, and WLM across 5 Gradient-Flow SDEs in Paired and Unpaired settings for train (first 10 marginals) and test (next 10 marginals).

| SDE | Method | Paired | | Unpaired | |
|---|---|---|---|---|---|
| | | Train $W_1$ ($\pm$SE) | Test $W_1$ ($\pm$SE) | Train $W_1$ ($\pm$SE) | Test $W_1$ ($\pm$SE) |
| Bohachevsky | JKONET* | $0.0814 \pm 0.0033$ | $0.0969 \pm 0.0018$ | $0.1969 \pm 0.0128$ | $0.3393 \pm 0.0149$ |
| | NN-APPEX | $0.0798 \pm 0.0021$ | $0.0917 \pm 0.0008$ | $0.0696 \pm 0.0029$ | $\mathbf{0.0742 \pm 0.0017}$ |
| | WLM (learnable friction) | $\mathbf{0.0388 \pm 0.0007}$ | $\mathbf{0.0488 \pm 0.0011}$ | $\mathbf{0.0660 \pm 0.0022}$ | $0.0907 \pm 0.0023$ |
| | WLM (0 friction) | $0.3195 \pm 0.0389$ | $0.3838 \pm 0.0460$ | $0.2690 \pm 0.0323$ | $0.3241 \pm 0.0262$ |
| Oakley O'Hagan | JKONET* | $0.0578 \pm 0.0025$ | $0.2067 \pm 0.0292$ | $0.1951 \pm 0.0340$ | $1.7709 \pm 0.3550$ |
| | NN-APPEX | $0.0526 \pm 0.0011$ | $0.0514 \pm 0.0018$ | $0.0729 \pm 0.0024$ | $0.1375 \pm 0.0179$ |
| | WLM (learnable friction) | $\mathbf{0.0415 \pm 0.0006}$ | $\mathbf{0.0509 \pm 0.0013}$ | $\mathbf{0.0430 \pm 0.0005}$ | $\mathbf{0.0734 \pm 0.0026}$ |
| | WLM (0 friction) | $0.0704 \pm 0.0062$ | $0.2462 \pm 0.0370$ | $0.0616 \pm 0.0032$ | $0.2470 \pm 0.0396$ |
| Quadratic | JKONET* | $0.0723 \pm 0.0046$ | $0.0753 \pm 0.0015$ | $0.2665 \pm 0.0415$ | $0.7161 \pm 0.0453$ |
| | NN-APPEX | $0.0432 \pm 0.0015$ | $\mathbf{0.0365 \pm 0.0005}$ | $0.0538 \pm 0.0019$ | $0.0631 \pm 0.0028$ |
| | WLM (learnable friction) | $\mathbf{0.0339 \pm 0.0008}$ | $0.0488 \pm 0.0021$ | $\mathbf{0.0391 \pm 0.0009}$ | $\mathbf{0.0555 \pm 0.0024}$ |
| | WLM (0 friction) | $0.0505 \pm 0.0021$ | $0.0964 \pm 0.0085$ | $0.0651 \pm 0.0035$ | $0.1089 \pm 0.0062$ |
| Styblinski-Tang | JKONET* | $0.1359 \pm 0.0186$ | $0.3791 \pm 0.0386$ | $0.2381 \pm 0.0409$ | $2.0202 \pm 0.3516$ |
| | NN-APPEX | $0.1142 \pm 0.0124$ | $\mathbf{0.2643 \pm 0.0206}$ | $0.2066 \pm 0.0243$ | $0.7183 \pm 0.0746$ |
| | WLM (learnable friction) | $0.1164 \pm 0.0134$ | $0.3983 \pm 0.0505$ | $0.1147 \pm 0.0132$ | $\mathbf{0.7091 \pm 0.0917}$ |
| | WLM (0 friction) | $\mathbf{0.0861 \pm 0.0075}$ | $0.3898 \pm 0.0724$ | $\mathbf{0.1134 \pm 0.0119}$ | $0.7271 \pm 0.1101$ |
| Wavy Plateau | JKONET* | $0.0759 \pm 0.0046$ | $0.2048 \pm 0.0293$ | $0.2827 \pm 0.0699$ | $3.2447 \pm 0.5395$ |
| | NN-APPEX | $0.1116 \pm 0.0127$ | $0.2128 \pm 0.0071$ | $0.1085 \pm 0.0064$ | $0.3090 \pm 0.0298$ |
| | WLM (learnable friction) | $0.0770 \pm 0.0064$ | $\mathbf{0.1383 \pm 0.0057}$ | $\mathbf{0.0750 \pm 0.0054}$ | $\mathbf{0.3002 \pm 0.0306}$ |
| | WLM (0 friction) | $\mathbf{0.0671 \pm 0.0039}$ | $0.1598 \pm 0.0186$ | $0.0879 \pm 0.0058$ | $0.3249 \pm 0.0415$ |

**Implementation of Baselines.** With one parameter exception, we use the suggested implementations for both SBIRR and SNAP-MMD for the oceans dataset used for interpolation and forecasting experiments in Shen et al. (2024) and Berlinghieri et al. (2025) respectively. Due to memory overflow, to implement the iterative proportion fitting SB-solver in SBIRR, we use 10 time substeps rather than 50 substeps. We note that the corresponding results are comparable and sometimes better than reported in the original papers (Shen et al., 2024; Berlinghieri et al., 2025).

For the action matching (Neklyudov et al., 2023a) and DICE (Blickhan et al., 2025) baselines on the oceans dataset, we use a 2-layer MLP with hidden dimension 64 and SiLU activations. We then ran each method using Adam with a constant learning rate of $10^{-4}$ for 2,000 iterations (AM, UAM, sAM) and 5000 iterations (DICE), with batch size 256 and gradient clipping at norm 1. For sAM we use diffusion coefficient $\sigma = 0.1$. We note that DICE (Blickhan et al., 2025) introduced a more stable action matching loss over discrete observations, which mitigates the over-optimization of a space-independent, but time-inhomogeneous potential. Despite this, we observe that DICE also diverges significantly during training, likely due to the sparsity of the data, which has only 111 particles per marginal, with large time gaps between observations. We

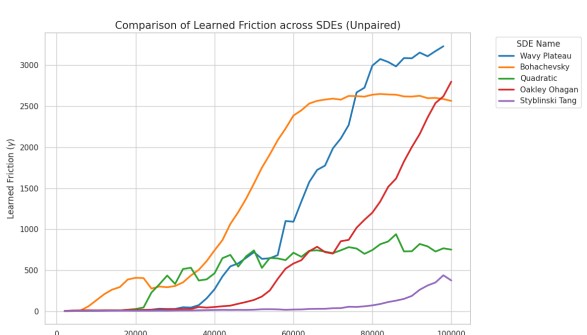

*(a)* Learnable friction for gradient flows (unpaired setting)

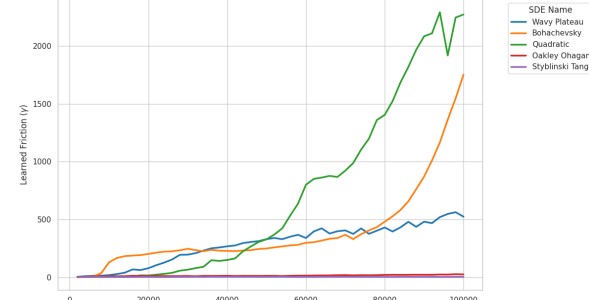

*(b)* Learnable friction for gradient flows (paired setting).

*Figure 6.* WLM Learned Friction ($\gamma$) curve for each SDEs over $100,000$ training epochs. Minimum learned friction at $100k$ epochs is $\gamma = 5.28$ (Styblinski-Tang paired setting), which collapses to the equilibrium distribution almost instantaneously.

therefore report DICE at its best checkpoint (epoch 3500) rather than at the end of training.

### C.3. Embryonic sc-RNA data

For the EB dataset (Moon et al., 2019), we follow standard pre-processing of 5 dimensional PCA and whitening the data, as first done in Tong et al. (2020), and subsequently replicated in many experiments (Neklyudov et al., 2023b; Kapusniak et al., 2024; Tong et al., 2023b). For leave-one-out interpolation, we train 3 separate models, using the remaining 4 marginals to predict the last one. We use the provided RNA velocity to simulate our rollout from $p_0$ during training, and to simulate from the previous observed marginal $p_{h-1}$ when interpolating the holdout, but we do not notice a significant difference between this choice and starting with $v_0 = 0$. Indeed, it has been conjectured that the velocity for this dataset may not be highly informative (Tong et al., 2020), and we also recover moderate to high dissipation, which places less emphasis on the importance of the initial velocities.

**WLM implementation.**   We modify the default architecture by dropping the feed forward inner dimension to 256. We also use weight decay $1e - 3$ and dropout 0.1. For the training loss, we use the Sinkhorn divergence loss with $p = 1$. For learnable friction, we initialize $\gamma = 1$ and friction learning rate $1e - 2$. We run for $10,000$ epochs and use an EMA of 0.999. We use mini-batching with size 1024 and report additional results for other batch sizes in Figure 8.

**Implementation of baselines.**   We implement the action matching (Neklyudov et al., 2023a) and DICE (Blickhan et al., 2025) baselines with the same implementation specified for the ocean vortex interpolation. We report DICE at its best checkpoint (epoch 3000) rather than at the end of training.

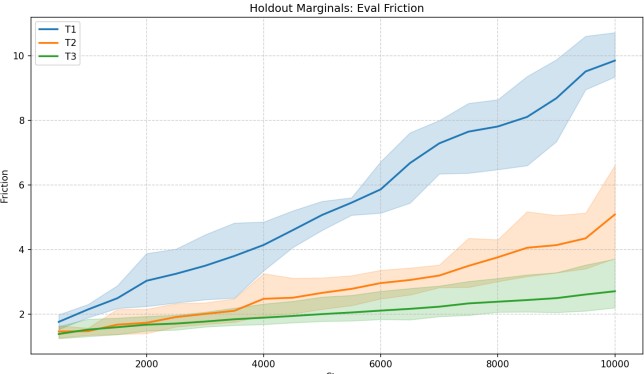

*Figure 7.* Learnable friction on the Embryoid Body (EB) dataset for different holdout times.

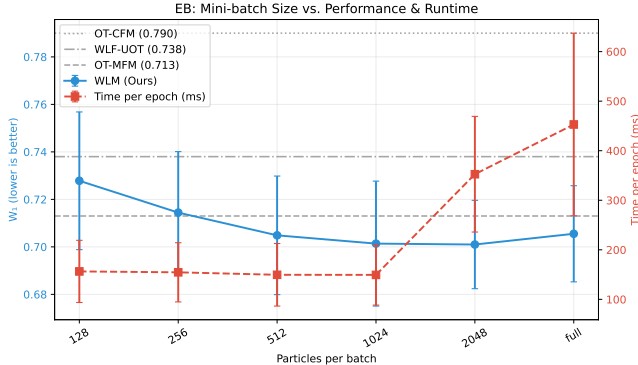

*Figure 8.* Runtime vs. performance when performing mini-batching with `WLM` averaged over 3 random seeds.

### C.4. Boids

We implement the Boids algorithm using a Vicsek-style interacting particle system based on the classic Boids algorithm. By default, we simulate $N = 1000$ agents in $\mathbb{R}^2$ adhering to local interaction rules, which is essentially the three Boids interaction rules with a boundary condition. In particular, at each time step $t$, the velocity is updated based on the state of its neighbours, and the acceleration is determined by the weighted sum of four distinct forces:

*Table 7.* Boids: Average $W_1$ distances between true and learned marginals for different initializations ($v_0$) on the same random seed. We distinguish between train times (first 50) and forecast (next 50).

| $v_0$ | Train $W_1$ ($\pm$SE) | Forecast $W_1$ ($\pm$SE) |
|---|---|---|
| True | $0.530 \pm 0.034$ | $1.523 \pm 0.051$ |
| DICE | $0.684 \pm 0.027$ | $1.521 \pm 0.044$ |
| Zero | $0.726 \pm 0.026$ | $1.645 \pm 0.052$ |

- Separation: short-range repulsion to avoid colliding wth flockmates. Agents move away from flockmates within an inner radius $R_{inner}$.

- Alignment: velocity matching with flockmates. Steer towards average velocity difference of all neighbours within an outer radius $R_{outer}$.

- Cohesion: Long-range attraction. Steer towards average relative position of flockmates within $R_{outer}$.

- Boundary: Restorative force applied when a Boid exceeds the simulation boundary, to steer back to origin.

Our default parameters are $R_{inner} = 0.3$, $R_{outer} = 1.0$, and different weights for enacted forces based on the four rules. By default, separation force is weighted $0.1$, alignment is weighted $0.3$, cohesion is weighted $0.005$, and boundary is weighted $0.5$, and triggered at magnitude $5.0$ away from the origin.

**WLM Implementation.** We use the default hyperparameters with $p = 1$ for the Sinkhorn divergence, and run for $30,000$ epochs.

**Implementation of Baselines.** We use default implementations for both JKONET$^*$ and NN-APPEX, as done for experiments on the same benchmark dataset. See (Terpin et al., 2024, Appendix C) and (Guan et al., 2026, Section 6) respectively.

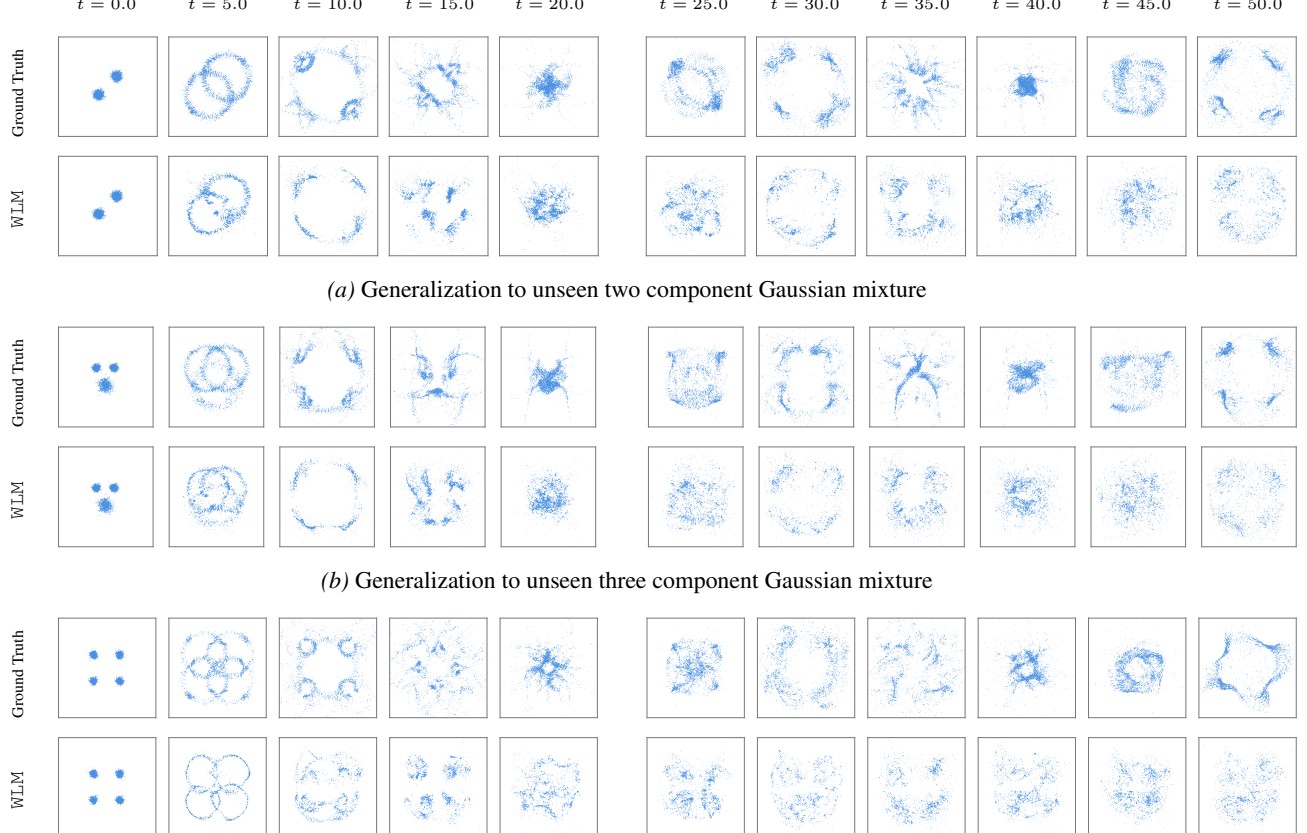

*(a)* Generalization to unseen two component Gaussian mixture

*(b)* Generalization to unseen three component Gaussian mixture

*(c)* Generalization to unseen four component Gaussian mixture

*Figure 9.* **Predicting Boids on unseen dynamics:** Qualitative comparison of ground truth Boids dynamics (top row of each panel) versus the predicted `WLM` dynamics (bottom row of each panel) for three unseen Gaussian mixture initial distributions. `WLM` was trained on 50 frames whose population was a centered Gaussian (see Figure 5).

