# OpenReview forum: "A Call to Lagrangian Action: Learning Population Mechanics from Temporal Snapshots"
_ICML.cc/2026/Conference — ICML 2026 spotlight_

### Official Review · Reviewer_dapy · 2026-03-11

**Soundness:** 3
**Presentation:** 3
**Significance:** 3
**Originality:** 3
**Overall Recommendation:** 5
**Confidence:** 4

**Summary:**

This paper proposes a method for learning continuous dynamics from sparse data snapshots. The authors first present the classical field theory formulation of the optimal transport problem, and then derive the more expressive WLM Framework by modifying the form of the Lagrangian. WLM uses a neural network to parameterize the variation of the potential energy function with respect to the probability density, thereby learning the dynamics from data. The paper validates the capabilities of WLM on multiple synthetic and real-world datasets, demonstrating that WLM achieves significant advantages in both interpolation and prediction tasks.

**Compliance With Llm Reviewing Policy:**

Affirmed.

**Final Justification:**

This paper derives the single-particle equation of motion from a classical field theory action (describing an ensemble of particles), aiming to learn the unknown potential energy functional from data to infer the dynamics of the particle ensemble. The mathematical derivations in this paper are solid. I initially gave it a score of 4 because I was unfamiliar with certain details of the algorithm, such as how to fit the potential energy functional using neural networks. The authors' rebuttal has addressed all my concerns, so I am raising my score to 5.

**Key Questions For Authors:**

Based on the weaknesses listed above, I would like to kindly ask the authors the following questions:

1. Could you please explain the rationale behind specifying the initial velocities as described in lines 271-274? If a known potential landscape is unavailable, or if the initial probability density $p_{0}(x)$ of the data cannot be accurately estimated, how should the initial velocities be set?
2. Furthermore, "learning least-action dynamics governed by an unknown action from data" is a challenging task because "learning the action" and "solving the dynamics" are inherently coupled. One could treat the currently learned dynamics as the ground truth to find an action, while simultaneously, an action directly dictates a specific dynamic. It appears to me that by specifying the initial velocities, this paper bypasses the learning of the least-action dynamics, thereby avoiding the adverse effects of the aforementioned action-dynamics coupling. I am curious if the authors intend to attempt learning the least-action dynamics? If so, how would this be approached? (e.g., by treating the initial velocities as the solution to an optimization problem?) Additionally, since the current study focuses on first-order dynamical systems, this implies that velocities can be discontinuous or re-specified at various time points. Would the initial velocities at all these individual time points need to be learned or solved for?
3. Since $\mathcal{U}[\rho]$ depends only on $\rho$, for an empirical measure composed of particles/data points, exchanging any two particles should leave $\mathcal{U}[\rho]$ unchanged. How does $\Psi _{\theta}(x^{(1)},x^{(2)},\cdots ,x^{(N)})$ guarantee this permutation invariance? Moreover, $\Psi _{\theta}$ takes the positions of all particles as input. When the number of particles is large and the data dimensionality is high, wouldn't this incur a massive computational burden? How can this be mitigated?

Please feel free to point out if any of my questions are misguided. I would be more than happy to raise my score if the authors could address these concerns.

**Limitations:**

yes

**Strengths And Weaknesses:**

### Strengths
1. By modifying the classical field theory formulation of optimal transport, this paper introduces the WLM Framework, derives the equations of motion (the continuity equation and the HJB equation) under the new Lagrangian, and utilizes the HJB equation to provide the material derivative of the velocity field. This allows for the direct simulation of particle trajectories starting from initial conditions. This approach makes excellent use of the mathematical structure of the problem.
2. The modified Lagrangian explicitly contains time, which allows the WLM framework to handle scenarios where the learning system exhibits energy dissipation or growth (e.g., active matter). Furthermore, the modified Lagrangian can incorporate potential energy functionals, making its form highly flexible. This enables the WLM framework to model various scenarios, including particle swarms governed by classical mechanics or the probability density of particles governed by the Schrödinger equation.

### Weaknesses
1. The paper simulates particle motion based on two ODEs, which requires the initial positions and initial velocities of the particles. However, the initial velocities are directly specified rather than obtained by solving an optimization problem.
2. The paper uses a parameterized $\Psi_{\theta} (x^{(1)}, x^{(2)},\cdots x^{(N)})$ to learn the underlying potential energy functional $\mathcal{U}[\rho]$ from the data. This neural network requires all data points as input, which might be overly complex. Additionally, this parameterization method may not guarantee that $\Psi_{\theta}$ adheres to the inherent properties of a potential energy functional.

---

> ### Author Rebuttal · Authors · 2026-03-31
>
> Thank you for your detailed feedback! We are glad that you appreciated the mathematical structure offered by WLM for modeling population dynamics, as well as the significant advantages that our method offered for both interpolation and forecasting. Below, we respond to your three key questions:
> > Q1 (i) Clarify L271-274. (ii) How does WLM handle unknown initial velocity?
>
> i) Recall that by Prop. 2.5, gradient flows can be characterized by a second order Hamiltonian system with infinite friction, and that this representation is stable for any choice of initial velocity (for details, see response T2 to reviewer ehux). Thus, our theory suggests that if friction is learnable, then we can specify any initial velocity, and explain the true dynamics with infinite friction. We thus chose $v_0 = 0$ for simplicity, but any initial velocity would be fine in principle. For the second choice of initial velocity, Prop 2.5 also showed that gradient flows can technically be represented by a conservative Hamiltonian system (no friction), but the caveat is that this is a highly unstable system that requires an initial velocity parallel to the gradient field $\nabla s_0$. For gradient-flow SDEs with drift $\nabla \Psi$ and diffusivity $\sigma^2$, this amounts to $\nabla s_t(x)=\nabla (\Psi(x) + \frac{\sigma^2}{2} \log p_t(x))$.
>
> ii) Great question! It is important to handle the case where the true initial velocity of the population is unknown, and must be estimated or misspecified. We first note that this is the case in two of our four main experiments: gradient flows in Sec. 4.1 (as explained above), and EB single cell in Sec. 4.3 (estimated RNA velocity, which Tong et al. (2020) notes to be particularly noisy for this dataset). To provide further evidence that WLM is robust to different estimates of the initial velocity, we ran an additional experiment on Boids, where the initial velocity is instead estimated from the data (we used DICE by Blickhan et al. (2025), effectively a more stable version of Action Matching by Neklyudov at al. (2023)) or completely misspecified as 0. We compared results to when WLM is given the true initial velocity on a new seed:
> |v₀|Train W1↓|Forecast W1↓|
> |---|---|---|
> |True|.530±.034|1.523±.051|
> |DICE|.684±.027|1.521±.044|
> |Zero|.726±.026|1.645±.052|
>
> The fitted and forecasted dynamics by WLM perform similarly well, despite different choices of initial velocity. We have also checked animations to confirm that qualitative performance is consistently strong.
> > Q2(i) Clarifying how dynamics, action, and $v_0$ are coupled, and approaches for inference. (ii) Do velocities at all times need to be specified?
>
> (i) You are right that “learning the action” (Lagrangian perspective) and “solving the dynamics” (Hamiltonian perspective) are coupled. In our paper, we primarily consider the setting where the initial velocity is also known, since we derived in Theorem 2.2 that action minimization induces a second order system, which is completely determined by the potential energy, the friction, and the initial velocity. WLM thus considers the first two quantities as learnable parameters, while assuming that the initial velocity is known in the data. Alternatively, we agree that one can do what you suggested and jointly optimize over all three of these physical parameters with a single optimizer. We hope that our answer in Q1(ii) sufficiently addresses concerns, by showing on multiple experiments that WLM is robust to velocity misspecification, and that it can be paired with velocity estimation methods as an initial subprocedure.
>
> (ii) The Hamiltonian dynamics is completely specified by the potential energy (together with the damping coefficient) and the initial momentum; therefore, only the velocity at the initial time is needed for simulation and training. We also clarify that WLM primarily considers second order dynamics, but can also model and learn first-order dynamics, e.g. gradient flows, which are independent of the initial velocity (overdamped limit).
> > Q3. i) How does our architecture guarantee permutation invariance between particles? ii) Clarification on the computational burden and how it can be mitigated.
>
> i) For this, please see response 3. to reviewer h7SA. There, we describe our self-attention architecture for parameterizing the potential energy (see also lines 860-870 of our Appendix), and explain why it has the desired invariance and expressivity properties. In particular, transformers process the population as an unordered set.
>
> ii) Computational complexity and mitigation strategies, e.g. Flash-attention and mini-batch rollouts, are also discussed in response 3 to h7SA. Even without these speed-ups, the runtimes were not prohibitive. However, we see a simulation-free method as important future work in order to make WLM more practical.
>
> Please let us know if any other points remain unclear, or if there are other aspects that you would like us to address.

---

> > ### Author Rebuttal · Reviewer_dapy · 2026-04-02
> >
> > I thank the authors for their thoughtful response. I still have a slight question regarding Q2 (ii), namely whether it is necessary to specify the initial velocity at each individual time point.
> >
> > Recalling the simpler case of classical mechanics, the dynamical equations of a system are also derived via variational principles in configuration space or phase space. Given the initial velocity (assuming the initial position is known), the system can evolve according to these dynamical equations. The difference here is that the Trajectory Inference problem discussed in this paper is a two-point boundary value problem (or a multi-point boundary value problem, as in Equation (3)).
> >
> > Still considering classical mechanics, for a two-point boundary value problem (given initial and final positions), I can solve for the initial velocity of the system based on the boundary conditions (specifically, solving for the time derivative of the generalized coordinates or the canonical momentum). If multiple successive time points $t_1, t_2, t_3, \dots, t_n$ are given, I can solve for the initial velocity at $t_1$ based on the boundary conditions at $t_1$ and $t_2$, solve for the initial velocity at $t_2$ based on the boundary conditions at $t_2$ and $t_3$, and so on. Therefore, I argued that the initial velocity at each time point needs to be re-specified. I would be interested to hear the authors' perspective on this.
> >
> > Overall, thanks again for the careful rebuttal. I believe the authors have profound expertise in theoretical physics, and I have benefited a lot from reading this paper. I will raise my score to 5.

---

> > > ### Author Response · Authors · 2026-04-06
> > >
> > > Thank you for the kind words. Here, we clarify that for both classical and population-level mechanics, the **continuous** motion is completely determined either by the initial position and velocity (Hamiltonian formalism) or by the initial and the final points (Lagrangian formalism). See corresponding chapters in, e.g. [Arnold et al, 1989]. However, both perspectives require defining the potential energy and the damping coefficient. For the trajectory inference problem (observed discrete snapshots), we do not have the potential energy defined. This creates room for pathological mechanics and “reward-hacking” of objectives, which, however, can be resolved by careful discretization of the continuous formulation (see Blickhan et al. (2025) for extensive discussion and examples).
> > >
> > > Thus, in our work, to avoid pathological examples, we focus on the continuous case (as derived in Theorem 2.1) and assume that the marginals are populated “densely enough” for learning the potential energy and the damping coefficient with the naive discretization techniques. Our experimental results show that this assumption holds up for these particular datasets, since WLM's learned dynamics are able to both interpolate and forecast unseen marginals. However, we believe that deriving a proper discretization of the continuous objectives as proposed in Blickhan et al. (2025) is an essential direction for future research.
> > >
> > > We have enjoyed discussing these theoretical points with you, and we hope that this answers your question.
> > >
> > > ## References
> > >
> > > Arnold, Vladimir Igorevich, Karen Vogtmann, and Alan Weinstein. Mathematical methods of classical mechanics. Vol. 60. New York: Springer, 1989.
> > >
> > > Blickhan, Tobias, et al. "DICE: Discrete inverse continuity equation for learning population dynamics." arXiv preprint arXiv:2507.05107 (2025).

---

### Official Review · Reviewer_dLXy · 2026-03-12

**Soundness:** 3
**Presentation:** 3
**Significance:** 3
**Originality:** 3
**Overall Recommendation:** 5
**Confidence:** 4

**Summary:**

The submission presents an approach to learn population dynamics from data. In contrast to existing methods, it considers non-gradient flow dynamics, allowing the model to represent oscillatory motion (on the population level). This is done by considering a more general class of Lagrangians on Wasserstein space (featuring a term $\exp \gamma t$ - $\gamma = 0$ reduces to the gradient flow case) and fitting it to data under the assumption that observed dynamics are optimal with respect to that Lagrangian in the sense of a Least Action Principle. The method is tested in numerical experiments; it performs well on data that can be reasonably approximated by gradient flows and, interestingly, best with $\gamma \neq 0$ in these cases. On data that is oscillating, it as expected outperforms these methods by a lot.

**Compliance With Llm Reviewing Policy:**

Affirmed.

**Final Justification:**

I will retain my score since the rebuttal phase has reinforced my initial assessment of the paper. It is relevant, well-presented, and novel. On the weakness side, the reviews agreed that "the tone of theoretical discovery is overstated". But this is something that can be addressed in the camera-ready version and the author's responses in the rebuttal phase have largely given the impression that this will happen.

**Key Questions For Authors:**

1) The interaction potential $\mathcal U \approx \Psi$ you learn seems to be full-batch. At this point, can I really expect speed-up over solving the full-order problem, when evaluation of $\Psi$ at all $(x_i)_{i=1}^N$ seeminly is $\mathcal O(N^2)$? In Table 4, instead of comparing to gradient-flow methods that certainly fail, a comparison to the full-order method would be more insightful.

2) For the experiments outlined in Appendix C.1, you recover $\gamma \gg 1$. Do you recover $\gamma = 0$ when $\sigma = 0$? In that case, the true dynamics are a gradient flow, so if I am not mistaken, this would be the expected result. $\gamma \approx 500$ is a remarkable value then given that the added noise is not very large $\sigma = 1$.

3) (minor) Is it really "intriguing" that WLM can describe the Schrödinger equation? It seems to be it is just re-deriving Bohmian mechanics, which fall exactly into the form of continuity equation (for $\partial_t \rho$) + Hamilton-Jacobi equation (for $\partial_t s$).

**Limitations:**

yes

**Strengths And Weaknesses:**

Soundness: This is a sound paper. The Theorems in this paper are formal. When contrasted to the (cited) works by Ambrosio/Gigli/Savare, it is clear that issues of regularity are swept under the rug. This is completely adequate for an ICML publication, but I consider most of the results more formal propositions and illustrative computations rather than theorems.

---

Presentation: The paper is well-written and easy to follow. The concept of Lagrangian Action on Wasserstein space is discussed in the (cited) 2023 paper by Neklyudov et al. and Villani's textbook features a full chapter (Optimal Transport: Old and New (2009), Chapter 7) on it. I think it could be more explicitly stated which part of this construction is taken from where and what is original.

---

Significance: Population dynamics inference is a topic of significant interest in the last years and the presented paper is a valuable addition.

---

Originality: As stated, I do not believe the concepts or theoretical results are necessarily new, but this is the first work I am aware of that attempts to simulate non-gradient population dynamics (besides the cited CurlyFM work, which is quite different in its approach and requires prior assumptions on the learned flow). See above under 'Presentation' regarding closely related literature.

---

> ### Author Rebuttal · Authors · 2026-03-31
>
> Thank you for reviewing our paper; we are glad that you agree that our paper is well written and that it contributes a valuable approach for learning population dynamics. We respond to your three questions below:
>
> > 1i) “The interaction potential  $\mathcal{U} \approx \Psi$ you learn seems to be full-batch. At this point, can I really expect speed-up over solving the full-order problem, when evaluation of $\Psi$ at all $(x_i)_{i=1}^N$ seeminly is $O(N^2)$? ii) In Table 4, instead of comparing to gradient-flow methods that certainly fail, a comparison to the full-order method would be more insightful.”
>
> i) Yes, using minibatches instead of the entire population during simulation rollouts can improve the computational efficiency of the method. Indeed, Prop 3.1 shows that we can exactly parameterize the population-level potential energy $\mathcal{U}[\rho]$ with a function that takes in the samples $\Psi(x^{(1)}, …, x^{(N)}$, which we by default parameterize with self-attention (see lines 860-870 in Appendix). As we discuss in response 3 to reviewer h7SA, mathematical theory for self-attention suggests that batching can be used to re-simulate the different subpopulations for more efficient training, since transformers process the population as an unordered set, and learn interaction weights that are independent of the population size $N$. For simplicity, and since the population sizes were moderate (in the thousands), we trained using the initial population of particles as a full batch in all experiments.
>
> ii) For the comparison with the methods modeling the potential that depends on all $N$ individuals in the population, we note that the baseline JKOnet* by Terpin et al. (2024) (stylized as “jkonet_star” in the docs https://jkonet-star.readthedocs.io/en/latest/pages/guide.html) does model full-order interaction, while inferring a gradient flow. Despite this, it failed to learn the Boids dynamics (Table 4).
> > 2. For the experiments outlined in Appendix C.1, you recover $\gamma >> 1$. Do you recover $\gamma = 0$ when $\sigma =0$? In that case, the true dynamics are a gradient flow, so if I am not mistaken, this would be the expected result. $\gamma \approx 500$ is a remarkable value then given that the added noise is not very large ($\sigma =1$) .
>
> We believe there is a minor misunderstanding here. First, stochasticity $\sigma$ does not change whether or not the dynamics are a gradient flow. Indeed, no matter the value of $\sigma \ge 0$, we can write the population dynamics as $\partial_t p_t = -\nabla \cdot (p_t \nabla s_t)$, where $s_t(x) = \Psi(x) + \frac{\sigma^2}{2} \log p_t(x)$. Thus, under the canonical formulation for continuous Wasserstein dynamics, each particle follows the gradient field $\nabla s_t$ (note the time-dependency of the function $s_t$). Second, to determine the recovered value of $\gamma$, we note that Prop 2.5 proves that the stable representation of a gradient flow (which holds for any initial velocity) corresponds to overdamped friction. Thus, no matter what stochasticity $\sigma$ is, our theory suggests that if friction is learnable, then, for a gradient flow, a very high value should be recovered given an arbitrary initial velocity, and this is indeed what we find. We will add a figure that provides the visual intuition for Prop 2.5 in a revision, which should help clarify this point.
>
> > (minor) Is it really "intriguing" that WLM can describe the Schrödinger equation?
>
> This is a good observation. Another reviewer pointed out the paper “Wasserstein Hamiltonian flows” by Chow et al. (2019), which performs the same derivation for the Schrödinger equation in their Example 3. In our revision, we will include this reference in addition to the reference by Neklyudov et al. (2024) that we already included for this fact, to more clearly indicate that this is a known result. See also response T1 to reviewer ehux for details on how we will revise the paper to present our theoretical contributions relative to existing literature.
>
> Please let us know if we addressed all your concerns, and please don’t hesitate to give us more suggestions that could improve your score/ our paper.
>
>
> # References
> Chow, Shui-Nee, Wuchen Li, and Haomin Zhou. "Wasserstein hamiltonian flows." Journal of Differential Equations 268.3 (2020): 1205-1219.
>
> Neklyudov, Kirill, et al. "A computational framework for solving Wasserstein Lagrangian flows." Proceedings of the 41st International Conference on Machine Learning. 2024.
>
> Terpin, Antonio, et al. "Learning diffusion at lightspeed." Advances in Neural Information Processing Systems 37 (2024): 6797-6832.

---

> > ### Author Rebuttal · Reviewer_dLXy · 2026-04-02
> >
> > Thank you for responding. I will retain my (high) score for this paper.
> >
> > 1) (i) I remain unconvinced about the computational speed-up (and the high training cost), but I do not consider this a deal-breaker. The $N^2$ complexity is a problem, since it competes with smarter neighbor-list book-keeping methods that scale like $k N$. See, for example, the boids code from JAX, M.D.: A Framework for Differentiable Physics (Neurips 2020), which is a citation that is worth adding. The code is available on https://github.com/jax-md/jax-md.
> > If the goal of this method is to learn a surrogate model, then the surrogate model should be faster than the full-order one, and I do not see this happening without some tricks. This does, however, leave the application of systems identification. (ii) As I said, I do not find it surprising that JKONet* cannot learn the boid dynamics (Figure 1 is a good illustration) and the insights I can gain from Table 4 are hence very limited.
> >
> > 2) Thank you for the clarification. My question was not well-phrased, what I meant is that an analytical expression to compare the learned potential is available in the case $\sigma = 0$.
> >
> > 3) "Wasserstein Hamiltonian flows” by Chow et al. (2019) is a very good addition, I like their presentation a lot. . I would like to once again suggest to reference Villani's textbook chapter on Lagrangian costs and clarify the novelty of contributions relative to that. The fact that gradient flows are the overdamped limit of Hamiltonian systems is well-known and is stated, for example, in that textbook on page 630 (the introduction to Chapter 23, "Gradient Flows I"). Overall, I agree with reviewer ehux that any effort made to clarify the novelty is well-spent. I think the theoretical contributions should be presented accurately, this includes my comment on the formal theorems (under "Soundness").

---

> > > ### Author Response · Authors · 2026-04-06
> > >
> > > Thank you for the continued feedback, we would like to follow up on each of these minor points in order to provide clarification, and to confirm how we will further improve the paper.
> > >
> > > 1(i): We would like to highlight that the goal of our proposed WLM method is not to learn a surrogate model to achieve faster simulation of the known dynamics. Rather, it is to learn the underlying laws from the observations. Although the JAX simulation of multi-particle dynamics is very relevant and we will definitely include it in the literature review, it is not directly comparable to our method, which infers the Boids dynamics purely from observations.
> > >
> > > To further study the computational cost of WLM, we ran an additional ablation over different mini-batch sizes, reporting performance vs. runtime trade-off offered by subsampling the initial population during rollout training. We consider the EB leave-one-out experiment, and for each population batch size, we re-ran the experiment with default hyperparameters (e.g. 100k epochs) on 3 seeds for each holdout index:
> > >
> > > | Particles per batch | W₁ | Runtime per epoch (ms) |
> > > |:---:|:---:|:---:|
> > > | 128 | 0.764 ± 0.022 | 131 ± 27 |
> > > | 256 | 0.756 ± 0.015 | 140 ± 44 |
> > > | 512 | 0.746 ± 0.026 | 138 ± 47 |
> > > | 1024 | 0.740 ± 0.019 | 142 ± 42 |
> > > | 2048 | 0.710 ± 0.017 | 332 ± 76 |
> > > | full (2381) | 0.690 ± 0.033 | 516 ± 147 |
> > >
> > > For consistency, the $W_1$ metric between the true holdout marginal and the predicted marginal is computed by simulating all samples from the previous marginal to the holdout marginal with the learned model (trained on mini-batches). We also plot the results against the three best performing previous methods as baselines in https://anonymous.4open.science/r/rebuttal_figures_2026-FB46/combined_w1_runtime.pdf. The results show that significant computational speed up (at slight performance cost) can indeed be obtained with mini-batching, since we roll out less particles at each epoch. With mini-batching, we can still outperform all previous methods with ~60% of the runtime budget. In fact, even with only ~25% of the runtime budget, WLM still beats all methods from previous papers except WLF by Neklyudov et al. (2023) and MFM by Kapuśniak et al. (2024).
> > >
> > > 1(ii): We emphasize that we view Boids as the most impressive and challenging experiment, due to its complexity from emergent dynamics, and the fact that we evaluate forecasting. Thus, while it may be unsurprising that gradient-flow methods cannot learn Boid dynamics, we were not aware of existing methods in the population dynamics literature that could serve as suitable baselines. We agree with your point that the qualitative figures offer more insights into the performance of the methods, and we also attach supplementary animations, e.g. we run the learned model on a completely different boids population here https://anonymous.4open.science/r/rebuttal_figures_2026-FB46/boids_animations/boids_generalization_to_unseen_population_1.gif and in other gifs in the same folder. The original training data (and WLM's forecasting performance) is also included in this folder.
> > >
> > > 2: Good point. The analytic expression is indeed more complicated with diffusivity $\sigma^2 > 0$ due to the score term $\frac{\sigma^2}{2} \nabla \log p_t(x)$. We also note that although the analytic formula is available, it cannot generally be used as a ground true vector field because the same population dynamics can admit multiple solutions with different vector fields, e.g. non gradient dynamics that produce the same marginals. To provide further qualitative evidence of WLM’s inference quality, we include gifs of WLM’s predicted rollouts for each SDE: https://anonymous.4open.science/r/rebuttal_figures_2026-FB46/gf_sde_gifs/forecast_bohachevsky.gif. In particular, the second half of the animations are in the forecasting regime, and illustrate that WLM’s learned potential energy aligns with the gradient field’s potential landscape.
> > >
> > > 3: Thank you, we will cite Villani's textbook to clarify relevant work and also physical intuition. We note that in that chapter, Villani motivates gradient flows via overdamped mechanics but does not derive the Wasserstein Hamiltonian formalism itself. This was done by Ambrosio and Gangbo (2008) and Chow et al. (2019), which in turn considered only the undamped conservative regime.
> > >
> > > Note: edits were made to attach links to animations and to discuss additional references.
> > >
> > > ## References
> > > Ambrosio, L., & Gangbo, W. (2008). Hamiltonian ODEs in the Wasserstein space of probability measures. Communications on Pure and Applied Mathematics: A Journal Issued by the Courant Institute of Mathematical Sciences, 61(1), 18-53.

---

### Official Review · Reviewer_h7SA · 2026-03-12

**Soundness:** 4
**Presentation:** 2
**Significance:** 3
**Originality:** 3
**Overall Recommendation:** 5
**Confidence:** 5

**Summary:**

The authors propose a learnable Hamiltonian system for modeling a distributional dynamics mapping between snapshot timepoints. The main innovation seems to be the ability to represent non-gradient dynamics (e.g. oscillations) without either resorting to a reference process (I assume like conditional probability paths in flow matching) or an unstructured, non-transport-like flow as in neural ODEs. The particular Hamiltonian they choose demonstrably shifts between Newtonian and other types of mechanics as a certain friction parameter is varied. Their framework, Wasserstein Lagrangian Mechanics, can be implemented in a neural network and trained by distributional losses over repeated rollouts. They also show how a hairy functional derivative can be computed easily with autograd. Despite not being identifiable except with additional assumptions, their Hamiltonian model performs better than baseline models on several data sets, including those with decidedly non-gradient-flow behaviors.

**Compliance With Llm Reviewing Policy:**

Affirmed.

**Key Questions For Authors:**

I have no questions per se which could improve my score, but my main concerns are stated above.

**Limitations:**

Yes

**Strengths And Weaknesses:**

Soundness: Fairly rigorous and good benchmarking. Solid range of data sets representing qualitatively different challenges.

Presentation: I would downplay some of the initial theory (see originality), put it in the appendix, and add more figures for the scRNAseq and whirlpool stuff: otherwise, it’s hard to get a qualitative sense of the performance, especially for the scRNAseq, where performance is competitive but not dramatically better. Also, I don’t think Figure 1 gets your point across. If the point is you can discover a Hamiltonian mechanics that explains a periodic dynamics or limit cycle, this should be shown explicitly. E.g. I was disappointed not to see a figure of the whirlpools.

Significance: I really like the goal of model discovery using a very general but still structured class of mechanics but which can nevertheless recapitulate cyclic and other non-gradient flows. The ability to extrapolate is also great, even if the approach isn’t simulation free.

Originality: I like the theoretical bent of the paper, but I do find apparent conceptual contributions to be slightly overstated. My understanding is that previous theoretical work, for instance by Gangbo, has already looked at these sort of second-order, Hamiltonian formulations in Wasserstein geometry. I think it will be important for the authors to revisit that mathematical literature to see if it is cited properly. To be clear,  the current results are still significant even if the main innovation is situating that earlier work in a learnable framework and with a particular form (e.g. with that damping term). But I do think the tone of theoretical discovery is overstated.

---

> ### Author Rebuttal · Authors · 2026-03-31
>
> Thank you for carefully reading our paper and for offering thoughtful feedback! We share the same assessment of the paper’s strengths, i.e. that it proposes a learnable framework for a very general but structured class of mechanics, and that the method’s “ability to extrapolate is great, even if the approach isn’t simulation free”. We reply to your main points below.
>
> > 1. _Figures for qualitative performance_: “Fairly rigorous and good benchmarking. Solid range of data sets representing qualitatively different challenges. (...) [I suggest to] add more figures for the scRNAseq and whirlpool stuff.”
>
> Great suggestion! We will complement our numerical results with figures of the learned dynamics for oceans and single-cell, in order to give qualitative evidence of WLM’s ability to learn diverse dynamics (see both at https://anonymous.4open.science/r/rebuttal_figures_2026-FB46/README.md). We will also add a figure that visualizes Prop. 2.5 (see response T2 to reviewer ehux).
>
> > 2. _Clarifying theoretical novelty_: “I like the theoretical bent of the paper, but I do find apparent conceptual contributions to be slightly overstated. (...) I think it will be important for the authors to revisit that mathematical literature to see if it is cited properly. To be clear, the current results are still significant even if the main innovation is situating that earlier work in a learnable framework and with a particular form (e.g. with that damping term). But I do think the tone of theoretical discovery is overstated.”
>
> Thank you for bringing relevant mathematical works to our attention. As discussed in response T1-T2 to reviewer ehux, we will prominently cite these works and better clarify our own theoretical contributions. After reviewing the literature, we agree that previous work has derived second-order Wasserstein Hamiltonian equations, with the most relevant work being “Wasserstein Hamiltonian flows” by Chow et al. (2019). Our Hamiltonian formulation from Theorem 2.2 offers greater generality than their Prop. 2 via the consideration of a damped Wasserstein Lagrangian, which is important for practical inference, but we will pare down the presentation of our Hamiltonian derivations as being completely original. To the best of our knowledge, our Prop. 2.5 remains novel, and we think it is particularly impactful for the ML community, since it shows that WLM encompasses gradient flows (and much more).
>
> >  3. _Computational complexity_: “I really like the goal of model discovery using a very general but still structured class of mechanics but which can nevertheless recapitulate cyclic and other non-gradient flows. The ability to extrapolate is also great, even if the approach isn’t simulation free.”
>
> We completely agree with your assessment of our paper’s strengths, and also that its biggest limitation is its computational complexity. A few reviewers asked questions about this. To simplify presentation, we provide a detailed discussion of WLM’s implementation, its computational complexity, its runtimes on main experiments, and ways to reduce runtime here:
>
> First, Prop 3.1 shows that we can exactly parameterize the population-level potential energy $\mathcal{U}[\rho]$ with a function $\Psi(x^{(1)}, …, x^{(N)})$ that takes in the samples, which we parameterize with self-attention (see lines 860-870 in Appendix). While self-attention classically requires heavier complexity on the order of $O(N^2d)$, it has become computationally efficient through Flash-attention (Dao et al. 2022), which provides speed-ups through improved memory access. The distributional loss during training is also computed rapidly with fast GPU kernel solvers (https://www.kernel-operations.io/geomloss/).
>
> The main computational bottleneck of the approach is the simulation of dynamics, which is why we noted simulation-free training as important future work. As established in recent literature viewing transformers as interacting particle systems (Geshkovski et al., 2025), self-attention acts as a consistent discretization of a population-level mean-field limit. This potentially allows for efficiency gains from minibatch rollouts. However, to reduce the number of confounding factors, we roll out the full initial population in all experiments, since the population sizes were moderate (in the thousands). We had noted in L876-879 that experiments ran between 5-30 hours. In fact, logs confirm that all main experiments finished within 15 hours, with most under 5 hours. Thus, while the computational load is more intense than baselines, we did not find it to be prohibitively expensive, even without some of the aforementioned speed-up tricks.
>
> # References
> Geshkovski, Borjan, et al. "A mathematical perspective on transformers." Bulletin of the American Mathematical Society 62.3 (2025): 427-479.
>
> Dao, Tri, et al. "Flashattention: Fast and memory-efficient exact attention with io-awareness." Advances in neural information processing systems 35 (2022): 16344-16359.

---

> > ### Author Rebuttal · Reviewer_h7SA · 2026-04-03
> >
> > I thank the authors for their very thoughtful reply. I think the MS will be greatly strengthened with the planned revisions, especially by situating their contribution in the broader mathematical literature and by clearer presentation of their framework's qualitative performance. I will maintain my score to reflect that I still believe this is a strong submission which deserves to be accepted.

---

> > > ### Author Response · Authors · 2026-04-06
> > >
> > > Thank you, we greatly appreciate the helpful feedback, and we also agree that the edits will improve the manuscript. Please note that to we also ran a supplementary ablation experiment over mini-batch size to explore the performance vs. runtime trade-off, which we will include in a revision https://anonymous.4open.science/r/rebuttal_figures_2026-FB46/combined_w1_runtime.pdf.

---

### Official Review · Reviewer_ehux · 2026-03-13

**Soundness:** 3
**Presentation:** 3
**Significance:** 3
**Originality:** 2
**Overall Recommendation:** 4
**Confidence:** 2

**Summary:**

The paper considers the problem of learning population dynamics from temporal snapshots. That is, we observe a dynamical system at time points $t_i$, where at each observation time $t_i$ we have access to a set of samples $[ x_{t_i}^{(j)} ]_{j=1}^N$ drawn from the time-marginal distribution $p_t$ (these samples are unaligned). Based on these samples, we wish to learn a dynamical system model that propagates a set of particles such that the marginal distribution of the particles at time $t$ matches $p_t$. To this end, the authors introduce a damped Wasserstein Lagrangian and derive the equations of motion (at the particle level) using Hamilton's principle of least action.

The resulting Hamiltonian dynamics depend on the potential energy (and damping factor). Hence, the learning problem boils down to learning a potential energy functional such that the induced dynamics match the given marginals. To this end, the authors propose to parameterize a neural network for mapping the population of $N$ particles to the potential energy evaluated at the empirical marginal distribution defined by those particles. This results in a neural ODE on the N-fold product state space. The learning involves rolling out this neural ODE over the observation time horizon and, at each observation time point $t_i$, computing a divergence between particle clouds that define the loss function.

**Compliance With Llm Reviewing Policy:**

Affirmed.

**Final Justification:**

See rebuttal acknowledgement

**Key Questions For Authors:**

**Theory**

* What parts of the theoretical developments and the establishment of the WLM formulation in Section 2 are novel? For instance, how does it relate to [1]? Since the theoretical foundation is a substantial part of the paper my score is based on the uncertainty that I feel about this point.

* I did not quite understand the details of Proposition 2.5. Could you please elaborate and provide an intuitive explanation of the two different interpretations studied in the proposition?

**Algorithm and evaluation**

* How do you compute the divergence D when training the algorithm? Is it also $W_1$ (I don't think this is specified?) and if yes, do you use Sinkhorn?
* Is it correct that the proposed algorithm assumes that you use the same number of particles $N$ as the number of training data points, or do you in fact use a different number? (I don't see any reason for why this should not be possible, but I might have missed something) Specifically, I'm asking because to me the algorithm seems to scale poorly with $N$ - both the fact that you need to compute the divergence between particle clouds at each time point and iteration, and due to the fact that the neural ODE operates on the $N$-fold product state space. Could you please comment on the complexity and training time of the algorithm (for some representative experiments)?
* To me action matching seems to be the most closely related method, but you are not comparing against this in the numerical evaluation, right? Why not? It should be applicable to the interpolation experiments, no?
* Is it a correct interpretation that your model is able to forecast because you learn a potential energy functional that is independent of $t$, whereas action matching learns a time-dependent action?
* I got a bit confused about the different baselines when reading the numerical evaluation. I could not see any references to SBIRR or SNAP-MMD (Section 4.2), nor explanations of what OT-CFM or MFM are. Is MFM (Table 2) the same as I-MFM or OT-MFM? What is Vanilla SB (only mentioned in table)?
* Do you actually learn curling dynamics in Sect 4.2? It's hard to determine what the particle-level dynamics look like by only considering the $W_1$ distance at the population level. Have you looked qualitatively at the learned dynamics?



[1] Shui-Nee Chow, Wuchen Li, Haomin Zhou
Wasserstein Hamiltonian flows, Journal of Differential Equations, 2020.
https://www.sciencedirect.com/science/article/pii/S0022039619303882

**Limitations:**

See above

**Strengths And Weaknesses:**

I enjoyed reading this paper and found it both interesting and well written. However, I am not expert on Wasserstein mechanics (and my comments need to be read in the light of this) and therefore I struggled to understand how novel the theoretical development is.

**Strengths**

* Well-written paper with a solid theoretical foundation
* The damped Wasserstein Lagrangian Mechanics (WLM) that underpin the proposed methods are general and encompass several related models. The authors show how classical mechanics, quantum mechanics, and Wasserstein gradient flows all can be seen as special cases of WLM.
* Strong empirical performance across several numerical examples.

**Weaknesses**

* It was not clear to me to what extent the theoretical development is new.
* While simple, the resulting algorithm appears to be very costly.
* Some potentially relevant baselines are missing.

---

> ### Author Rebuttal · Authors · 2026-03-31
>
> Thank you for reviewing our paper! We are glad that you enjoyed reading the paper, and that you appreciated the generality of our theory, as well as our method’s strong performance across datasets. Below, we respond to each of your questions pertaining to the theory (which we label T1-T2) and to the algorithm (which we label A1-A6).
>
> > T1: “What parts of the theoretical developments and the establishment of the WLM formulation in Section 2 are novel? For instance, how does it relate to [1]?”
>
> The main novelty of the proposed framework, in particular Theorem 2.2, is the derivation of the Hamiltonian equations from a **damped** Wasserstein Lagrangian, which thus incorporates dissipation into the population mechanics. Importantly, Prop. 2.5 then builds on this by noting that gradient flows, the de facto model for learning population dynamics in the ML community, admit a stable second-order representation under overdamped friction. In our revision, we will clarify that similar derivations for the second-order population-level Hamiltonian equations have been done in the mathematical literature [Chow et al. (2019), Prop. 2] for the undamped time-homogeneous Wasserstein Lagrangian formulation. Thank you for bringing this work to our attention!
>
>
> > T2: Provide intuition for Prop. 2.5
>
> Prop. 2.5 proves that Wasserstein gradient flows (which are first-order dynamics) have two representations as second-order Wasserstein mechanics. First, gradient flows can be represented as a physical system sliding down the gradient landscape (in Wasserstein space) of a free energy functional $\mathcal{F}$, such that its velocity is determined by the landscape $-\nabla \mathcal{F}$. For this, we need to match the potential energy in WLM with the free energy $\mathcal{U}=\mathcal{F}$ and take the limit of overdamped friction. Importantly, this representation is stable: for *any initial velocity*, the overdamped system immediately self-corrects the dynamics to follow the gradient landscape. For the second perspective, we instead invert the free energy landscape, such that the gradient dynamics are instead viewed as the second-order motion of a system rolling up this _frictionless_ inverted landscape, and settling at the top, i.e. the free energy minimizer. The caveat is that this representation is unstable since it requires a precise initial velocity. We will add an illustrative figure to the camera-ready version pending acceptance.
>
> > A1-A2: Clarifying algorithm implementation: (i) Does the algorithm assume that the same number of particles is shared across time points? (ii) How is the divergence $\mathcal{D}$ computed during training? (iii) “Could you please comment on the complexity and training time of the algorithm?”
>
> (i)-(ii): In general, the proposed algorithm does not require the same number of particles across different timestamps. In practice, however, WLM performs rollouts from the first marginal, so we by default use the number of particles present in $p_0$. For settings like the EB experiment, where the number of samples varies, the divergence between the predicted and actual marginal $\mathcal{D}(\hat{p}_t, p_t)$ is between measures with differing sample sizes. This computation is handled by the Sinkhorn divergence from geomloss (https://www.kernel-operations.io/geomloss/) with p=2, as specified in “WLM default hyperparameters” in Appendix C. We did not find that changing the divergence (ex. p=1, or MMD) significantly changed performance.
>
> (iii): We agree that WLM's main limitation is its computational complexity. Please see response 3 to reviewer h7SA for a detailed discussion about WLM’s complexity, its runtimes on main experiments, and ways to reduce runtime.
>
> > A3-A4: The reviewer points out that action matching (AM) is a relevant method for interpolation, and asks for clarification that AM cannot forecast.
>
> We agree that AM is a relevant method. AM learns (time-varying) first-order gradient-flow dynamics, while WLM learns second-order Hamiltonian dynamics. You are right that AM is unable to forecast since it must parameterize time. We ran three AM variants (classic, unbalanced, entropic) as baselines for the interpolation experiments:
>
> # Oceans
>
> |W1↓|t2|t4|t6|t8|
> |---|---|---|---|---|
> |AM|.360|.480|.595|.458|
> |uAM|.340|.439|.597|.502|
> |eAM|.355|.171|.286|.217|
>
> # EB
>
> |M|W1↓|
> |---|---|
> |AM|.958±.101|
> |uAM|1.034±.176|
> |eAM|1.132±.065|
>
> AM performs similarly to other methods on EB (but not as well as WLM), and relatively worse on the curling ocean vortex currents, which is unsurprising since AM models gradient-flow dynamics.
>
> > A5: Missing references and details about baselines
>
> Thank you for pointing out missing citations! We will also overview baselines in the appendix.
>
> > A6: “Do you actually learn curling dynamics in Sect 4.2?”
>
> Yes! We will include a figure of WLM’s learned curling dynamics from the ocean vortex, see https://anonymous.4open.science/r/rebuttal_figures_2026-FB46/oceans_interpolation_figure.pdf

---

> > ### Author Rebuttal · Reviewer_ehux · 2026-04-02
> >
> > Thank you for the detailed reply. My main question about originality of the theoretical development has been addressed and I will increase my score to 4. I  but I still feel that the question about computational complexity should receive more attention in the paper. For instance, it would be interesting to see how sensitive the method is to the number of simulated particles (which could be smaller than the number of observed particles, if I understood your reply correctly) in the numerical experiments.

---

> > > ### Author Response · Authors · 2026-04-06
> > >
> > > We are happy to see that our rebuttal addressed your questions regarding our theoretical contributions. Here, we comprehensively address the remaining request about computational complexity by
> > >
> > > (1) Reporting the full breakdown of WLM’s runtimes per experiment,
> > >
> > > (2) Running an additional ablation over different batch sizes (for subsampling the initial population) in order to explore the efficiency vs performance trade-off.
> > >
> > > Both of these analyses will be included in the paper revision.
> > >
> > > For (1), we report the average runtime per experiment below, along with the number of particles present at the initial time, the number of time marginals during training, and the number of training epochs.
> > > | Experiment | Particles | Marginals | Epochs | Hours |
> > > |:---|---:|---:|---:|---:|
> > > | GF SDEs | 1000 | 9 | 100,000 | 9.22 (±2.07) |
> > > | Boids | 1000 | 49 | 30,000 | 13.52 (±1.20) |
> > > | Oceans (small) | 111 | 4 | 50,000 | 3.15 (±0.70) |
> > > | Oceans (big) | 400 | 9 | 50,000 | 4.05 (±0.46) |
> > > | EB | 2381 | 3 | 100,000 | 11.62 (±3.19) |
> > >
> > > For (2), we investigate the effectiveness of mini-batching on the EB leave-one-out experiment and report the performance vs. runtime trade-off below. For each population batch size (i.e. number of particles subsampled from the initial time), we re-ran the experiment with default hyperparameters (e.g. 100k epochs) on 3 seeds for each holdout index.
> > > | Particles per batch | W₁ | Runtime per epoch (ms) |
> > > |:---:|:---:|:---:|
> > > | 128 | 0.764 ± 0.022 | 131 ± 27 |
> > > | 256 | 0.756 ± 0.015 | 140 ± 44 |
> > > | 512 | 0.746 ± 0.026 | 138 ± 47 |
> > > | 1024 | 0.740 ± 0.019 | 142 ± 42 |
> > > | 2048 | 0.710 ± 0.017 | 332 ± 76 |
> > > | full (2381) | 0.690 ± 0.033 | 516 ± 147 |
> > >
> > > For consistency, the $W_1$ metric between the true holdout marginal and the predicted marginal is computed by simulating all samples from the previous marginal to the holdout marginal with the learned model (trained on mini-batches). We also plot the results against the three best performing previous methods as baselines in https://anonymous.4open.science/r/rebuttal_figures_2026-FB46/combined_w1_runtime.pdf. The results thus show that significant computational speed up (at slight performance cost) can indeed be obtained with mini-batching, since we roll out less particles at each epoch. With mini-batching, we can still outperform all previous methods with ~60% of the runtime budget. In fact, even with only ~25% of the runtime budget, WLM still beats all methods from previous papers except WLF by Neklyudov et al. (2023) and MFM by Kapuśniak et al. (2024).
> > >
> > > We hope that these additional analyses for the main experiment runtimes and for mini-batching as a mitigation strategy address your remaining concerns about our manuscript’s discussion of computation complexity.

---

### Decision · Program_Chairs · 2026-04-30

**Decision:**

Accept (spotlight)

**Comment:**

This paper makes a strong contribution to the learning of population dynamics by introducing a more expressive Wasserstein Lagrangian mechanics framework capable of capturing non-gradient behaviors. The submission combines a compelling conceptual advance with solid technical development and strong empirical results across both synthetic and real datasets, consistently improving over gradient-flow and flow-matching-style baselines, particularly in settings where non-gradient dynamics are important. Reviewers broadly agreed that the paper is well written, technically sound, and significant, and the rebuttal successfully clarified the main concerns, especially regarding novelty. Overall, this is a strong paper and should be of broad interest to the community working on scientific machine learning, optimal transport, and population dynamics.